# Multi-Heteroatom Doped Fe@CN Activation Peroxomonosulfate for the Removal of Trace Organic Contaminants from Water: Optimizing Fabrication and Performance

**Jiamin Chen, Ruijun Ren, Yatao Liu, Chen Li, Zhenbei Wang and Fei Qi ***

Beijing Key Lab for Source Control Technology of Water Pollution, College of Environmental Science and Engineering, Beijing Forestry University, Beijing 100083, China; jmchen_2020@163.com (J.C.); renruijun526@126.com (R.R.); liuyatao233@163.com (Y.L.); li_c21@bjfu.edu.cn (C.L.); wangzhenbei119@163.com (Z.W.)
* Correspondence: qifei@bjfu.edu.cn

**Abstract:** Modification of catalysts by multi-heteroatom doping (S, P, B) is an effective way to improve the peroxomonosulfate activation performance of catalysts. In recent years, highly toxic and persistent trace organic contaminants have been frequently detected in water. Consequently, we proposed the advanced oxidation processes of peroxomonosulfate activated by multi-heteroatom doped Fe@CN (X-Fe@CN) to eliminate trace organic contaminants. The physical phases of X-Fe@CN and its precursors were characterized by X-ray diffraction and scanning electron microscopy. In evaluating the catalytic properties and iron ion leaching of X-Fe@CN-activated PMS for the removal of dicamba and atenolol, B-Fe@CN and PB-Fe@CN were selected and optimized. The active sites of the catalysts were characterized by X-ray photoelectron spectroscopy and Raman. The pathways of PMS activation by B-Fe@CN and PB-Fe@CN were identified in combination with electron paramagnetic resonance and electrochemical experiments. Defects, O-B-O and pyrrolic nitrogen on the surface of B-Fe@CN could adsorb and activate PMS to produce $SO_4^{\bullet-}$, $\cdot OH$ and $^1O_2$. Further doping with P enhanced the electron transfer on the catalyst surface, thus accelerating the activation of peroxomonosulfate. This study compared the effects of multi-heteroatom modifications and further demonstrated the synergistic effect between P and B, which can provide a theoretical basis for the selection of multi-heteroatom doped catalysts in water treatment.

**Keywords:** peroxymonosulfate; trace organic contaminants; multi-heteroatom doping; reaction mechanism

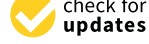



## 1. Introduction

Due to the substantial use of synthetic chemicals in urban areas, trace organic contaminants (TOrCs) have been consistently found in urban wastewater in recent years [1]. TOrCs can be moved downstream with the water during rainfall events or seep into groundwater, potentially leading to their presence in drinking water as their original form or as harmful transformation product [2,3]. It is imperative to address this issue and prevent the spread of TOrCs. Developing efficient water treatment technologies for TOrCs, such as pesticides and PPCPs (pharmaceuticals and personal care products), is crucial in reducing their presence within water bodies, mitigating environmental risks, and upholding the quality of reclaimed water [4,5].

Current wastewater treatment plants mainly use anaerobic and aerobic units; however, it is difficult to eliminate TOrCs by conventional wastewater treatment methods. These TOrCs, which are toxic, recalcitrant, and resistant to biological or physicochemical treatments, can be effectively removed using sulfate radical advanced oxidation processes (SR-AOPs) [6,7]. In SR-AOPs, TOrCs undergo oxidative removal by a variety of reactive

oxygen species (ROS) including $SO_4^{\bullet-}$, $\cdot OH$, superoxide radicals ($O_2^{\bullet-}$), and singlet oxygen ($^1O_2$). $SO_4^{\bullet-}$, which is highly electron accepting, can readily break the benzene ring via electron transfer and ultimately remove TOrCs into small molecules [8].

Transition metal oxides are the most effective catalysts for activation persulfates; however, the leaching of toxic metal ions poses a serious issue that restricts their development [9]. Among the alternatives, transition metal nitrogen–carbon materials fabricated from metal–organic frameworks (MOFs) materials as precursors exhibit high efficiency and low metal ions leaching due to their N coordination anchoring metal atom [10–13]. At the same time, materials such as carbon nitride are used as carriers to prepare uniformly dispersed materials, which is beneficial to reduce metal clusters [14]. Materials with small-sized metal clusters dispersed on carbon nitride can expose more metal active sites, such as those formed by Fe-N and other transition metals coordinated to N [12]. The surrounding carbon can enhance the interaction between the contaminant and the catalyst, which on the one hand can improve the catalytic performance of the material and on the other reduce the leaching of metal ions [15].

Heteroatom doping is commonly utilized to boost the catalytic activity of carbon-based materials [16]. Doping introduces diverse electronegativities into the carbon structure, which breaks the inherent inertia of the $sp^2$ hybridized carbon network, thereby amplifying the electron transfer rate and catalytic activity [17]. The atomic radii of nitrogen (N = 0.75 Å) were similar to those of carbon (C = 0.77 Å); however, their higher electronegativity is conducive to productive alterations in the electronic behavior of the carbon lattice [18].

Compared to N-doped carbon catalysts, incorporating two or more heteroatoms allows the interaction between the heteroatoms in the carbon catalysts. This leads to a disruption of carbon lattice symmetry, increasing the number of surface-active sites, and enlarges surface defects, ultimately enhancing the potential catalytic activity of the carbon catalysts. The inclusion of graphite N with other dopants (B, S, P, Se, or Si) has been demonstrated to enhance the interaction between peroxymonosulfate (PMS) and carbon catalysts, owing to an increase in spin and charge density [16]. Co-doping of phosphorus and boron on carbon catalysts improves their electronic conductivity [19]. In addition, the presence of phosphorus changes the p-type properties of boron to n-type doping, thereby increasing the spin density around the dopant and decreasing the band gap of carbon catalysts [19]. Li et al. found that co-doping of nitrogen, phosphorus, and sulfur significantly enhances the activity of carbon catalysts for oxygen reduction reaction [20]. The doping of heteroatoms also can transform catalytic removal pathways of catalysts [16]. For instance, P and N co-doping has been proven to increase surface defects in porous carbon materials, facilitating conversion of superoxide radicals to single-linear oxygen during PMS activation [21]. The possible competition for doping sites on the carbonaceous lattice and interactions between multiple heteroatoms result in a shift in the activation mechanism of the catalyst for refractor organics removal [22].

In this study, atenolol (ATL) and dicamba (DIC)) were selected to evaluate the effectiveness of PMS activation of X-Fe@CN. DIC is a hydrophilic herbicide that can easily dissolve in water, leading to its migration in agricultural ecosystems and ultimately contaminating soil and groundwater, thereby posing risks to water safety and human health [23]. β-blockers are a drug class primarily employed in the treatment of cardiovascular illnesses, including coronary heart disease and hypertension. ATL is a widespread β-blocker, regularly discovered in various environmental samples, including wastewater, surface water, groundwater, soil, and sediments, with levels reaching up to 300 μg·L$^{-1}$ [24]. To achieve efficient removal of these TOrCs, the study aims at optimizing the selection of the most appropriate multi-heteroatom doped Fe@CN, optimized Fe@CN fabricated parameters and its activation PMS' reaction parameters, analyzing the surface-active sites and active species produced, and uncovering the mechanism of TOrCs removal by X-Fe@CN activating PMS.

## 2. Materials and Methods

### 2.1. Chemicals and Reagents

Oxone ($2KHSO_5 \cdot KHSO_4 \cdot K_2SO_4$) was purchased from Alfa (Shanghai, China). Iron (III) chloride hexahydrate ($FeCl_3 \cdot 6H_2O$) and 2-aminoterephthalic acid ($NH_2$-BDC) (98%) were obtained from Aladdin (Shanghai, China). Ethanol absolute was obtained from Energy Chemical (Anqing, China). Melamine ($C_3H_6N_6$) (99%), thiourea ($CH_4N_2S$), boric acid ($H_3BO_3$) (99.5%), dicamba (DIC), atenolol (ATL), ammonium acetate ($CH_3COONH_4$), phosphoric acid ($H_3PO_4$) (85%), 5,5-dimethyl-1-pyrroline N-oxide (DMPO), and 2,2,6,6-ter amethylpiperidine (TEMP) were obtained from J&K (Beijing, China). High-performance liquid chromatography (HPLC)-grade methanol and N,N-dimethylformamide (DMF) were purchased from Macklin (Shanghai, China). Wahaha®purified water (Beijing, China) was used in all experiments. Chemicals used in this study were all analytical grade and used directly without purification.

### 2.2. Synthesis of Heteroatom-Doped Catalysts

#### 2.2.1. Synthesis of X-C₃N₄

Graphitic carbon nitride was synthesized via pyrolysis using muffle furnace by selecting melamine as raw material. The yellowish solid compound graphitic carbon nitride ($g$-$C_3N_4$) was obtained by adding 10.0 g of melamine to a ceramic crucible and calcining continuously for 4.0 h after heating to 550 °C at a ramp rate of 5 °C/min. Similarly, graphitic carbon nitrides co-doped with S, P, and B were successfully synthesized by a process of in situ thermal copolymerization of melamine with thiourea, diammonium dihydrogen phosphate [25–27], and boric acid as the dopants ($X$-$C_3N_4$, X including S, P, B, SP, SB, PB, SPB). The feedstock contents of different heteroatom-doped carbon nitride and the corresponding calcination parameters are demonstrated in Table S1.

#### 2.2.2. Synthesis of X-C₃N₄@NH₂-MIL-53(Fe) and X-Fe@CN

$FeCl_3$-$6H_2O$ (0.674 g) and 2-aminoterephthalic acid (0.453 g) were added to 56.0 mL DMF and stirred for more than 20 min until complete dissolution. Then, the obtained $X$-$C_3N_4$ (0.25 g) was added, stirred well, and then dispersed in an ultrasonic machine for 30 min [28]. The mixture was placed in a 100 mL reactor for a hydrothermal reaction at 170 °C for 12.0 h. After the reactor was cooled down to room temperature, the brown crystals obtained from the reaction were poured out, washed with DMF until the washing solution was nearly colorless, and then continued to be cleaned with anhydrous ethanol. After washing, the precursor ($NH_2$-MIL-53(Fe)@$X$-$C_3N_4$) was dried in a vacuum drying oven at 100 °C for 10.0 h. The precursor was finally pyrolyzed in a tube furnace under $N_2$ atmosphere (300 ± 5 mL/min) at an elevated temperature rate of 5 °C/min to 650 °C for 3.0 h, resulting in X-Fe@CN black powder. A diagram depicting the process of preparing X-Fe@CN is presented in Figure 1, while Table 1 shows pyrolysis products of precursors with $X$-$C_3N_4$@$NH_2$-MIL-53(Fe) and the corresponding names of obtained catalysts.

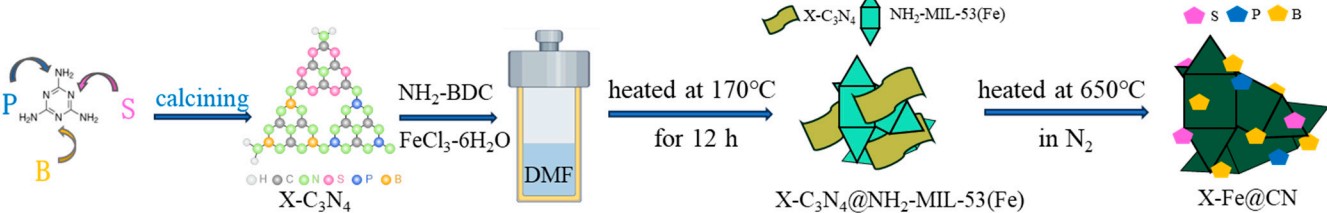

**Figure 1.** Scheme of the synthesis process of X-Fe@CN.

**Table 1.** Pyrolysis products of precursors with X-C$_3$N$_4$@NH$_2$-MIL-53(Fe) and corresponding names of obtained catalysts.

| Simple | Name |
|---|---|
| NH$_2$-MIL-53(Fe) | FexCN-650 |
| g-C$_3$N$_4$@NH$_2$-MIL-53(Fe) | Fe@CN |
| S-C$_3$N$_4$@NH$_2$-MIL-53(Fe) | S-Fe@CN |
| P-C$_3$N$_4$@NH$_2$-MIL-53(Fe) | P-Fe@CN |
| B-C$_3$N$_4$@NH$_2$-MIL-53(Fe) | B-Fe@CN |
| SP-C$_3$N$_4$@NH$_2$-MIL-53(Fe) | SP-Fe@CN |
| SB-C$_3$N$_4$@NH$_2$-MIL-53(Fe) | SB-Fe@CN |
| PB-C$_3$N$_4$@NH$_2$-MIL-53(Fe) | PB-Fe@CN |
| SPB-C$_3$N$_4$@NH$_2$-MIL-53(Fe) | SPB-Fe@CN |

### 2.3. Experimental Procedures

TOrCs removal performance of nine kinds of the above-obtained heteroatom-doped Fe@CN was evaluated using two kinds of TOrCs, DIC and ATL. The specific experimental procedure was as follows: the desired doses of X-Fe@CN and PMS were added simultaneously to a solution containing 200 mL of DIC (8.84 mg/L or ATL (10.65 mg/L), and the pH was adjusted to 7.0 by a NaOH solution (1.0 mol/L). The reaction was stirred using a magnetic stirrer, and the solution was extracted at specific time intervals (0–60 min) and immediately terminated with a Na$_2$S$_2$O$_3$ solution (100 mmol/L) to terminate the reaction. The samples obtained were stored in a refrigerator for subsequent assays. The detailed experimental procedures for the EPR experiment and the electrochemical experiment are described in Text S1.

### 2.4. Characterization and Analysis Methods

The morphological and structural properties of all fabricated X-Fe@CNs and their precursors were observed using scanning electron microscopy (SEM, Hitachi SU8010, Kyoto, Japan). The crystal structures of the resulting samples were characterized using Cu/Kα-rays ($\lambda$ = 1.5406 A, 2$\theta$ = 5–85°) on an X-ray powder diffraction analyzer (XRD, Shimadzu 7000, Kyoto, Japan). Raman spectra were obtained by a Raman Spectrum Analyzer (Raman, Horiba LabRAM Odyssey, Palaiseau, France) in the range of 1000–2000 cm$^{-1}$ with an excitation wavelength of 532 nm. X-ray photoelectron spectroscopy (XPS, ESCALAB MK-2, Thermo Scientific, Waltham, MA, USA) was used to analyze the elemental composition of the material and the chemical morphology of the surface. Electrochemical measurements such as cyclic voltammetry (CV), electrochemical impedance spectroscopy (EIS) and Tafel curves (TFL) were performed on an electrochemical workstation (CHI760E, Shanghai, China) equipped with a conventional three-electrode system. DIC (V(methanol):V(0.1% phosphoric acid) = 70:30) and ATL (V(methanol):V(10 mM ammonium acetate) = 40:60) were determined at 230 nm and 224 nm using high-performance liquid chromatography (HPLC, Waters E2695, Milford, MA, USA). Divalent and trivalent iron ions dissolved in the solution were determined using the o-phenanthroline colorimetric method. The main active species were monitored using an electron paramagnetic resonance spectrometer (EPR, Bruker EMXplus-10/1, Billerica, MA, USA). All abbreviations and their full names are listed in Table S2.

## 3. Results and Discussion

### 3.1. Characterization of X-Fe@CNs and Their Precursors

#### 3.1.1. XRD and SEM of X-C$_3$N$_4$

The crystal structure of X-C$_3$N$_4$ was characterized by XRD; it exhibits two distinct peaks that typify layered polymerized carbon nitride (Figure 2a). The peak (100) crystal face of g-C$_3$N$_4$ at the diffraction angle of 2$\theta$ = 13.21° represents the structural stacking of an in-plane conjugated aromatic ring in the lattice plane, whereas the latter peak (002) at 2$\theta$ = 27.6° is due to the tri-stacking of the $sp^2$ hybridization orbitals of the triazine moiety.

No additional peaks were detected, indicating the product's high purity and the presence of heteroatoms doped into the skeleton [26]. It was shown that the intensity of the two peaks in S--, P, and B-doped carbon nitride is reduced, suggesting that the samples have thinner nanosheets and poorer order of planar structural units [27]. Layer spacing *d* of the layered carbon nitride materials was deduced from Bragg's formula (Equation (1)) [29], which corresponds to (100) and (002) as 0.669 nm and 0.323 nm. The layered structure and layer spacing of the carbon nitride can be inferred from the position and intensity of the (002) peak in the XRD spectra, and the increase in layer spacing *d* is conducive to the exposure of more active sites, which further promotes PMS activation.

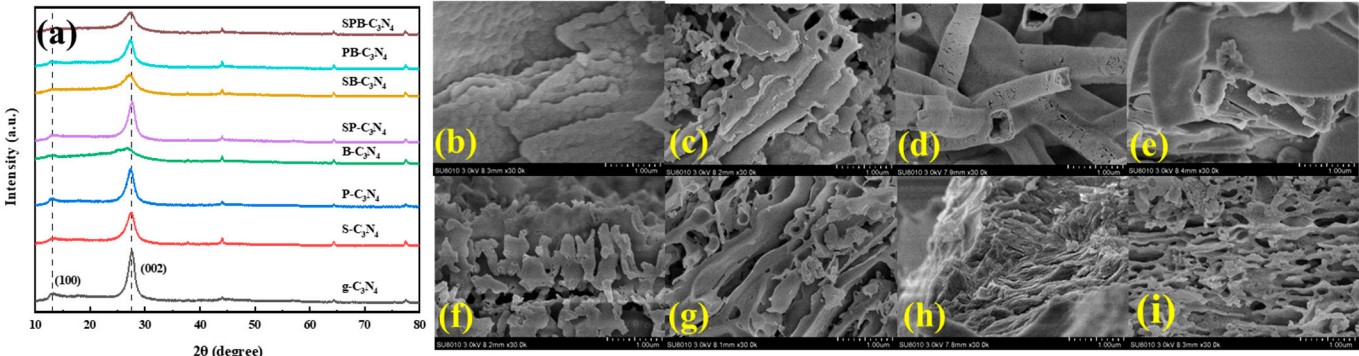

**Figure 2.** (**a**) XRD pattern of X-C$_3$N$_4$ and SEM images of (**b**) g-C$_3$N$_4$; (**c**) S-C$_3$N$_4$; (**d**) P-C$_3$N$_4$; (**e**) B-C$_3$N$_4$; (**f**) SP-C$_3$N$_4$; (**g**) SB-C$_3$N$_4$; (**h**) PB-C$_3$N$_4$; (**i**) SPB-C$_3$N$_4$.

According to Table S3, it can be seen that the (002) peak of B-C$_3$N$_4$ is slightly shifted from 27.6 to 26.8, and layer spacing *d* is increased to 0.331 nm. The peak position of g-C$_3$N$_4$ exhibits a smaller blue shift after S/P atom doping, while co-doping with B atoms leads to a greater blue shift. This implies that the interplanar stacking distance of a conjugated aromatic structure increases after doping with S (1.02 Å), P (1.10 Å), and B (0.82 Å) atoms, which showed larger atomic sizes than C (0.77 Å) and N (0.75 Å) [26]. The intensities of diffraction peaks at both the (002) crystal plane and the (100) crystal plane were significantly reduced after B doping, indicating that the polymerization degree of B-C$_3$N$_4$ was significantly higher than that of g-C$_3$N$_4$. Whereas S/P doping showed less effect on the polymerization degree of g-C$_3$N$_4$, the degree of effect increased after co-doping with B atoms. When the polymerization degree is higher, the interlayer hydrogen bonding density is relatively lower; therefore, the interlayer hydrogen bonding density of g-C$_3$N$_4$ decreased significantly after B doping [30]. The B atoms are more likely to be embedded in the lattice sites, causing lattice distortion [31]. The interlayer distances are enlarged after doping with heteroatoms in planar g-C$_3$N$_4$, which is in agreement with the literature, suggesting the creation of highly exfoliated X-C3N4 and a reduction in the planar size of the graphitic carbon layer [32,33].

$$2d\sin\theta = n\lambda. \tag{1}$$

The morphology of X-C$_3$N$_4$ was observed using SEM (Figure 2). The original g-C$_3$N$_4$ presented a typical aromatic laminar structure in a low adsorption aggregation morphology consisting of dense and thick nanosheets (Figure 2b). After S doping, S-C$_3$N$_4$ showed a porous lamellar structure in the plane (Figure 2c), and a large number of rough pores could be caused by the sublimation of S in the heat treatment. The morphology of P-C$_3$N$_4$ formed a hollow nanotube-like structure (Figure 2d), which is in agreement with what has been reported in the literature by Liu et al. [34]. B-C$_3$N$_4$ formed a layered structure that was readily exfoliated. B doping increased the interlayer spacing of the carbon nitride, which together with the XRD analysis could explain the significant decrease in the interlayer hydrogen bonding density, resulting in the formation of highly exfoliated carbon nitride. SP-doped g-C$_3$N$_4$ has a shoot-like or angular structure (Figure 2f), which may be attributed to the deformation of the tubular structure formed by P doping due to sulfur doping. The

irregular stacked structure of SB-C$_3$N$_4$ with holes retains the porous lamellar structure to a greater extent, but the lamellar structure produces deformed bending. PB-C$_3$N$_4$ forms a dense lamellar structure compared to B-C$_3$N$_4$, which corresponds to the spacing of the layers. In contrast, SPB-C$_3$N$_4$ exhibited a porous lamellar structure resembling that of SP-doped g-C$_3$N$_4$. It shared similarities with the morphology of SP-C$_3$N$_4$ but was looser and more porous. This could potentially be attributed to the reduced density of interlayer hydrogen bonding in g-C$_3$N$_4$ after B doping. Above all, the doping of heteroatoms can significantly affect the morphology of carbon nitride, potentially providing more active sites for redox reactions [35], which lays the foundation for subsequent optimization of X-Fe@CN.

### 3.1.2. XRD and SEM of X-C$_3$N$_4$@NH$_2$-MIL-53(Fe)

The crystal structure of X-C$_3$N$_4$@NH$_2$-MIL-53(Fe) has been characterized by XRD (Figure 3a). NH$_2$-MIL-53(Fe) prepared in this study showed distinct characteristic peaks at 8.9, 10.23, 16.7, 17.8, and 21.1°, which is consistent with the characteristic peaks of NH$_2$-MIL-53(Fe) in the literature [36], indicating the successful preparation of the material. Exogenous N doping was achieved by coupling with NH$_2$-MIL-53(Fe) through the addition of g-C$_3$N$_4$ according to Liu et al. [34]. g-C$_3$N$_4$@NH$_2$-MIL-53(Fe) contains the characteristic peak of NH$_2$-MIL-53(Fe) and a carbon peak around 27°, which is in agreement with the pattern reported by He et al. [37]. This indicates the successful coupling of NH$_2$-MIL-53(Fe) with g-C$_3$N$_4$ and the fact that they have successfully formed heterojunctions between the internal components [14].

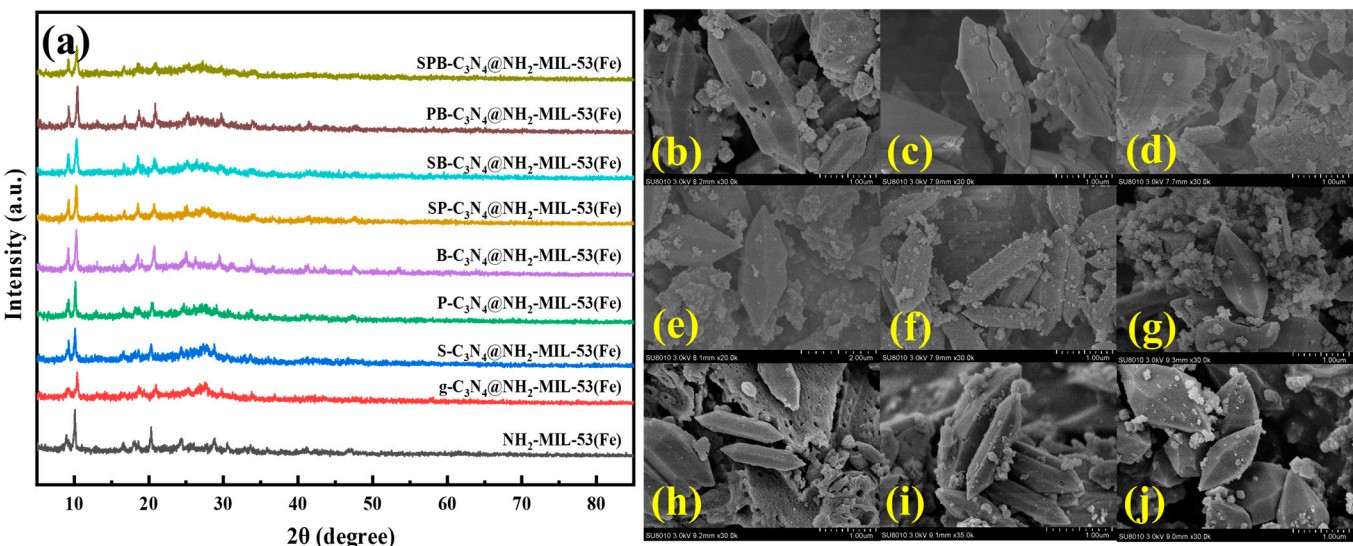

**Figure 3.** (**a**) XRD pattern of X-C$_3$N$_4$@NH$_2$-MIL-53(Fe) and SEM images of (**b**) NH$_2$-MIL-53(Fe); (**c**) g-C$_3$N$_4$@NH$_2$-MIL-53(Fe); (**d**) S-C$_3$N$_4$@NH$_2$-MIL-53(Fe); (**e**) P-C$_3$N$_4$@NH$_2$-MIL-53(Fe); (**f**) B-C$_3$N$_4$@NH$_2$-MIL-53(Fe); (**g**) SP-C$_3$N$_4$@NH$_2$-MIL-53(Fe); (**h**) SB-C$_3$N$_4$@NH$_2$-MIL-53(Fe); (**i**) PB-C$_3$N$_4$@NH$_2$-MIL-53(Fe); (**j**) SPB-C$_3$N$_4$@NH$_2$-MIL-53(Fe).

B- and PB-doped g-C$_3$N$_4$ formed composites with stronger peaks and increased crystallinity after complexing with NH$_2$-MIL-53(Fe), indicating that B and PB doping favors the formation of crystal particles. S-C$_3$N$_4$@NH$_2$-MIL-53(Fe) had a more pronounced carbon peak around 27°, and it has been shown that S doping can convert defective carbon into g-C$_3$N$_4$. Overall, X-C$_3$N$_4$@NH$_2$-MIL-53(Fe) has weak g-C$_3$N$_4$ and strong NH$_2$-MIL-53(Fe) peaks, indicating that the heteroatom-doped g-C$_3$N$_4$ and the NH$_2$-MIL-53(Fe) composite successfully formed a more regular structural framework.

The morphology of X-C$_3$N$_4$@NH$_2$-MIL-53(Fe) was characterized by SEM as shown in Figure 3. It can be seen that NH$_2$-MIL-53(Fe) forms a 2–3 μm spindle-type morphology (Figure 3b), and there are also some irregular spherical structures with particle sizes around

200 nm. g-C$_3$N$_4$@NH$_2$-MIL-53(Fe) with the addition of g-C$_3$N$_4$ retained the spindle-shaped morphology, and the crystal structure was not significantly modified. The spindle-shaped NH$_2$-MIL-53(Fe) closely adheres to the surface of the lamellar g-C$_3$N$_4$, which can change the morphology of g-C$_3$N$_4$ and increase its specific surface area.

Precursor S-C$_3$N$_4$@NH$_2$-MIL-53(Fe) with the addition of S-C$_3$N$_4$ retains the spindle-shaped morphology, and the spindle particles become smaller in size (800 nm–1.0 μm) and more elongated, but the morphology is more complete. After the addition of P-C$_3$N$_4$, SP-C$_3$N$_4$ and SPB-C$_3$N$_4$, the composites formed a wider spindle-type morphology, with particle sizes around 1.0–2.0 μm. The composites, after the addition of B-C$_3$N$_4$, SB-C$_3$N$_4$, and PB-C$_3$N$_4$, formed a regular narrow and long spindle-shaped morphology with a particle size of about 2.0 μm. Spindle morphology formed by the material doped with a P element was changed, the crystal aspect ratio was close to 2:1, there were more unformed particles, and the doping of the S element mainly affected the size of the spindle. XRD diffraction peak intensity was also higher in the B-doped element, indicating that B doping favors the formation of a more regular spindle morphology with a higher aspect ratio (4:1) in NH$_2$-MIL-53(Fe).

### 3.1.3. XRD and SEM of X-Fe@CN

As shown in Figure 4a, the pyrolyzed FexCN-650 and Fe@CN are in agreement with the references [34], indicating the successful fabrication of these two materials. Fe$^0$, α-Fe$_2$O$_3$, γ-Fe$_2$O$_3$ and Fe$_3$O$_4$ were the main physical phases of the pyrolysis products of FexCN-650. The diffraction peaks of α-Fe$_2$O$_3$ corresponding to Fe@CN disappeared, and the diffraction peaks of Fe$^0$ were significantly enhanced, which indicated that the large introduction of the externally added nitrogen favored the conversion of iron oxide clusters to Fe$^0$.

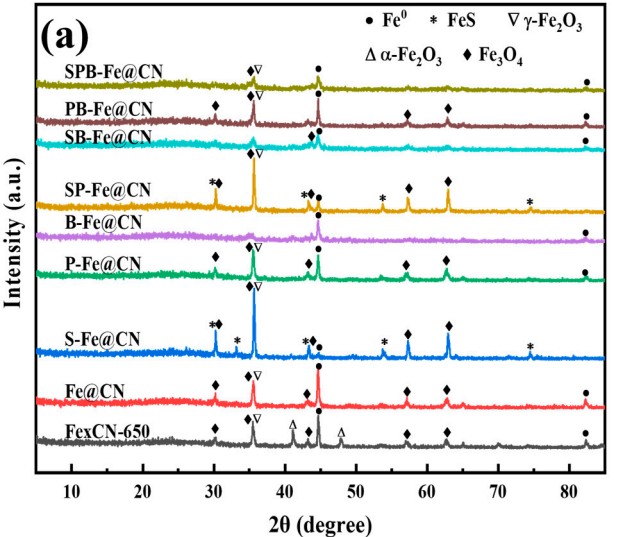
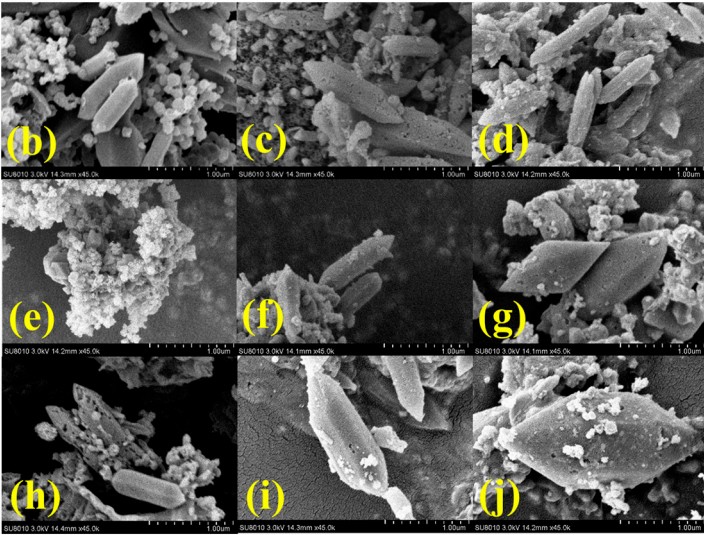

**Figure 4.** (**a**) XRD pattern of X-Fe@CN and SEM images of (**b**) FexCN-650; (**c**) Fe@CN; (**d**) S-Fe@CN; (**e**) P-Fe@CN; (**f**) B-Fe@CN; (**g**) SP-Fe@CN; (**h**) SB-Fe@CN; (**i**) PB-Fe@CN; (**j**) SPB-Fe@CN.

S-Fe@CN also shows FeS in addition to Fe$^0$, γ-Fe$_2$O$_3$ and Fe$_3$O$_4$. The peaks at 2θ = 30.0°, 33.7°, and 53.7° correspond to (100), (002), and (110) of the FeS crystal plane (JCPDS 37-0477) [38], respectively, confirming the successful doping of S. The peak intensity of Fe$^0$ is drastically reduced, and on the contrary, the diffraction peaks of γ-Fe$_2$O$_3$ and Fe$_3$O$_4$ are enhanced, probably due to the substitution of Fe sites by S monomers. The sublimation of sulfur monomers introduces more defects. These defects may expose encapsulated Fe$^0$ within the framework, and consequently lead to further oxidation. P-Fe@CN shows less change compared to Fe@CN, with a slight decrease in the intensity of the Fe$^0$ peak and a slight increase in the intensity of the Fe$_3$O$_4$ peak. The peak intensities of Fe$^0$, Fe$_3$O$_4$ and Fe$_2$O$_3$ in B-Fe@CN were significantly reduced, and the decrease in diffraction intensity

can be attributed to the lattice defects in Fe@CN due to B atoms on the one hand and the increase in the thickness of the catalyst on the surface of the metal particles on the other hand. The XRD of SP co-doping was similar to that of the S alone doping one, but diffraction peak intensity decreased, indicating that S doping exerted the main influence, while P doping reduced the diffraction peak intensity to a small extent. SB-Fe@CN showed similar diffraction peaks to B-Fe@CN, but the overall decrease in diffraction peak intensity indicated that B doping had a major influence, while further doping of S decreased the diffraction peak intensity, and the degree of crystallization of the material decreased when the two were co-doped. The significant decrease in $Fe_2O_3$ and $Fe_3O_4$ contents obtained from XRD can be attributed to the fact that S broke the Fe-O bond. P and B co-doping shows similar diffraction peaks to P-Fe@CN, and SPB triple co-doping is similar to B-Fe@CN. Since S is more electronegative with respect to P and B, it is more likely to displace $Fe^0$ when co-doped, resulting in a decrease in the intensity of the $Fe^0$ diffraction peak for both S/SP/SB/SPB doped materials.

The morphology of X-Fe@CN was characterized by SEM as shown in Figure 4. The actual percentage of element for X-Fe@CN was determined using EDS and is supplemented in Table S4. Based on EDS results, we determined that the sulfur content of the catalysts with the addition of S was in the range of 0.9–1.3 wt.%, the P content of the catalysts with the addition of P was in the range of 1.2–2.3 wt.%, and the B content of the catalysts with the addition of B was in the range of 0.1–0.78 wt.%. Since the material needs to be calcined twice and washed several times, most of the C, N, O, H and S elements were depleted in it, so the content of S was decreased and the contents of P and B were elevated compared to the mass percentage of the original predicted addition (Table S5). FexCN-650 maintained the spindle morphology of precursor $NH_2$-MIL-53(Fe) after pyrolysis (Figure 4b). Introduction of externally added nitrogen source g-$C_3N_4$ effectively prevents the aggregation of pyrolysis products and significantly improves the dispersion of the fusiform particle on the carbon layer [37]. Introduction of externally added nitrogen source g-$C_3N_4$ also leads to a better retention of morphology of the precursor during the pyrolysis process (Figure 4c). The spindle morphology of Fe@CN is consistent with that reported in the literature, further indicating its successful preparation [28].

The precursor structure of P-Fe@CN under high-temperature pyrolysis underwent significant collapse, forming irregular agglomerates and not retaining the complete spindle structure (Figure 4e). Most studies indicate that P doping increases the roughness of the material surface, while its original structure remains intact [19,39,40]. Conversely, P-Fe@CN morphology is disrupted, which could be linked to the P doping method and pyrolysis temperature. The obvious holes in the structure of SB-Fe@CN indicated that the spindle structure of its precursor is not stable enough, and it easily damaged and collapsed under high-temperature pyrolysis, which is consistent with the lower diffraction peak intensity of XRD (Figure 4h). The rest of X-Fe@CN better retained the spindle shape of the precursor, indicating that the doping of other heteroatoms does not significantly affect the stability of the spindle. The doping of heteroatoms had a significant effect on the crystal composition and morphology of the catalysts, which further affected the performance of X-Fe@CN activating PMS.

### 3.2. Removal Performance of TOrCs by X-Fe@CN Activating PMS

The catalytic performance of X-Fe@CN for PMS activation was assessed by exam-ining the removal efficiency of two contaminants, DIC and ATL (Figure 5). The removal of DIC and ATL followed pseudo-first-order kinetics, and the apparent reaction rate constant $k_{obs}$ was obtained from Equation (1) [41]. It was indicated that PB-Fe@CN demonstrated the highest $k_{obs}$ for DIC removal as the best activator, followed by S-Fe@CN, B-Fe@CN, SPB-Fe@CN, SP-Fe@CN, Fe@CN, P-Fe@CN, and SB-Fe@CN, in descending order. The catalytic activity of Fe@CN was significantly enhanced by inclusion of both internal and external nitrogen sources compared to FexCN-650 with only internally doped nitrogen sources resulting in a 29.4% to 40.0% increase in removal efficiency of the contaminant. However, P-

Fe@CN and SB-Fe@CN displayed decreased removal performance, while other X-Fe@CNs exhibited a slight enhancement. PB-Fe@CN exhibited significant efficiency in activating PMS for DIC removal, resulting in a 59.32% removal efficiency and an apparent reaction rate constant of 0.013 min$^{-1}$. $k_{obs}$ of ATL removal was ranked according to a decreasing order as follows: PB-Fe@CN > S-Fe@CN > B-Fe@CN > SP-Fe@CN > SPB-Fe@CN > Fe@CN > SB-Fe@CN > P-Fe@CN > FexCN-650 (Figure 5b). Among them, B-Fe@CN demonstrated the highest removal efficiency of 74.36%, with $k_{obs}$ increasing from 0.0099 min$^{-1}$ to 0.0408 min$^{-1}$ (Figure 5c). B-Fe@CN and PB-Fe@CN displayed superior performance in removing ATL, indicating a possible correlation between the nature of the contaminant (such as electrostatic attraction between PMS and ATL) and their removal performance. The removal of DIC using PMS activated by a PB-Fe@CN catalyst yielded a 59.32% reduction compared to the 62.76% achieved with ATL. The above reaction was finished at pH = 7.0, indicating that PMS was negatively charged, ATL was positively charged and DIC was negatively charged, with $p$ka(ATL) = 9.6, $p$ka(DIC) = 1.87, and $p$ka$^2$(PMS) = 9.4 [42]. ATL was found to be electrostatically attracted to PMS, while DIC was repelled, which could explain the enhanced removal of ATL by X-Fe@CN in comparison to DIC.

$$\ln(\frac{C_t}{C_0}) = -k_{obs}t. \tag{2}$$

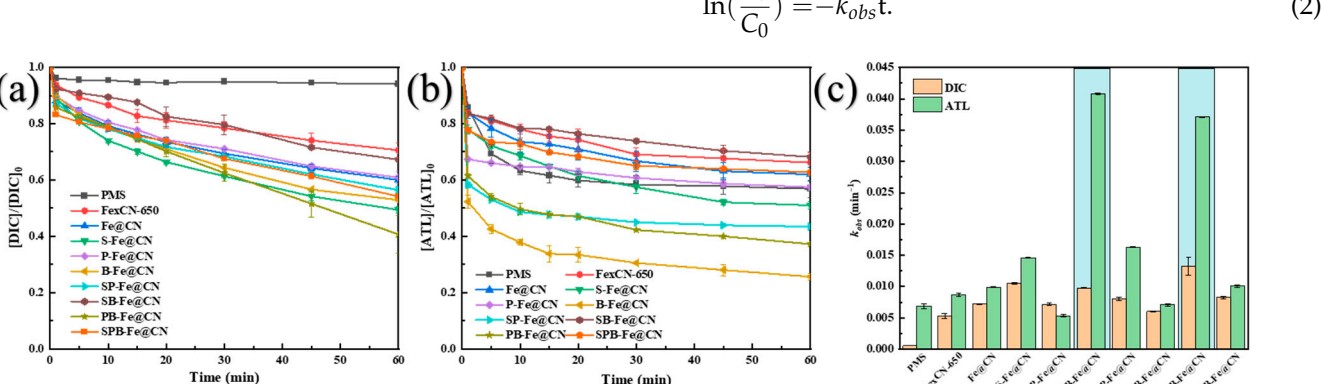

**Figure 5.** Removal efficiency of (**a**) DIC and (**b**) ATL in different reactions within 60 min; (**c**) $k_{obs}$ of DIC or ATL removal in different reactions. Experimental conditions: (PMS) = 1.30 mmol/L, (DIC) = 8.84 mg/L, (ATL) = 10.65 mg/L, (X-Fe@CN) = 0.1 g/L, pH = 7.0.

For ATL removal, X-Fe@CN doped with different heteroatoms exhibits similar performance differences as for DIC removal. Outstanding catalytic activity was observed among S-alone, B-doped, and PB-co-doped Fe@CN. As can be seen from Figure 6, there is a significant relationship between the catalytic performance of X-Fe@CN and its morphology. S-Fe@CN exhibited improved spindle morphology, and although the crystallization of Fe$^0$ decreased, FeS and Fe$_3$O$_4$ content increased due to oxidation of S. The increase in FeS and Fe$_3$O$_4$ leads to an increase in the active sites of Fe$^{2+}$, which further activates PMS [38]. Furthermore, the introduction of S brings about a large number of C-S-C (thiophene sulfur) groups [43]. According to the literature, thiophene sulfur is the primary active site of S-doped iron–carbon materials [44]. Since S has an extra electron compared to P, it is of interest due to its ability to displace carbon atoms and its strong synergistic impact with nitrogen dopants. Sulfur-induced structural defects yield a positive impact on charge dislocations and oxygen adsorption [43]. In contrast, the interactions between the doped carbon and molecular oxygen are influenced by the contribution of the two lone pairs of electrons from sulfur atoms. B-Fe@CN maintains its complete spindle morphology and a high level of zero-valent iron crystallization. PB-Fe@CN exhibited an exceptional ability to remove both DIC and ATL. The addition of P and B had a synergistic effect, where its doping retained the morphology of the material while the introduction of B and P atoms created numerous low-coordination active sites. The active sites introduced when S and B were doped individually were most likely responsible for the enhanced properties

of S-Fe@CN and B-Fe@CN. The co-doping of P and B showed synergistic effects on the removal performance.

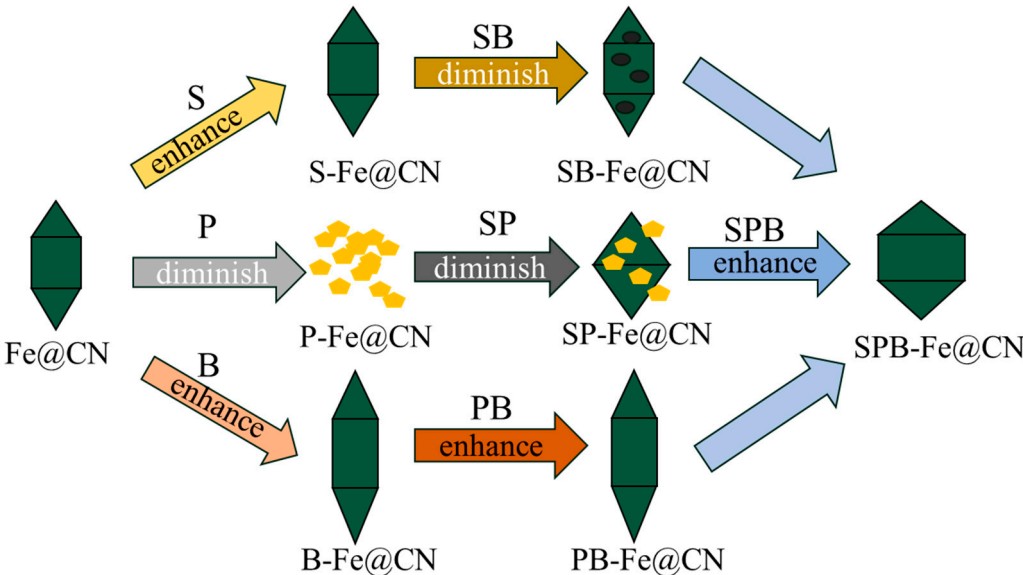

**Figure 6.** Schematic diagram of X-Fe@CN activating PMS versus morphology.

The co-doping of S and P into Fe@CN catalysts resulted in properties similar to those of Fe@CN doped solely with S, with no significant synergistic effect between the two. However, P doping caused a significant morphological change in SP-Fe@CNs, transforming them from spindle to tetragonal cones (Figure 6). The performance of Fe@CN after SPB co-doping was significantly reduced compared to PB-Fe@CN and was similar to S-Fe@CN, suggesting that the introduction of S impeded the synergistic effect of P and B. Preliminary results suggest that S and B doping has a negative impact on the performance of Fe@CN. Additionally, our findings indicate that P doping results in a significant decrease in catalytic activity. This observation may be attributed to the fact that P-doped materials did not form the characteristic spindle structure. This can be explained by the larger size and slightly smaller electronegativity of P atoms compared to C. P doping leads to an exaggeration of the graphite layer spacing and a redistribution of charges, thereby affecting the formation of the spindle structure. S and B co-doping, conversely, resulted in a reduction in catalytic activity, attributed to the presence of holes caused by the spindle structure collapse of SB-Fe@CN (Figure 6). Additionally, it is evident from Figure 4a that the medium SB-Fe@CN experienced a significant decrease in $Fe^0$, $Fe_2O_3$ and $Fe_3O_4$ content, further contributing to the loss of active X-Fe@CN sites. In summary, it can be seen that the maintenance of the spindle structure is the key to ensure the better catalytic activity of X-Fe@CN, and the spindle structure with higher aspect ratio exhibits superior catalytic activity.

### 3.3. Safety Assessment of X-Fe@CN

When utilizing X-Fe@CN activating PMS for DIC and ATL removal from water, it is crucial to uphold long-term evaluation of catalytic efficiency and application safety. To achieve this, it is imperative to ensure a stable structure while minimizing iron ion leaching. The dissolution of iron ion in the reaction is primarily attributed to persulfate being a weakly acidic oxidant, the chemical reactions between $Fe^0$ and $Fe^{2+}$ in PMS activation by X-Fe@CN [12]. Furthermore, a portion of the iron particles may be detached through physical means such as friction. The dissolved iron ions from all fabricated X-Fe@CN are depicted in Figure 7a–c. The majority of the iron ions that dissolved were trivalent. The dissolved iron ions from all fabricated X-Fe@CN were less than 3.0 mg/L after 60 min of reaction. The aforementioned results suggest that both $Fe^0$ and iron ions on X-Fe@CN participate in PMS activation. However, the doping of distinct heteroatoms introduces

varied species and amounts of iron involved in the reaction, ultimately influencing the safety and stability of the catalysts.

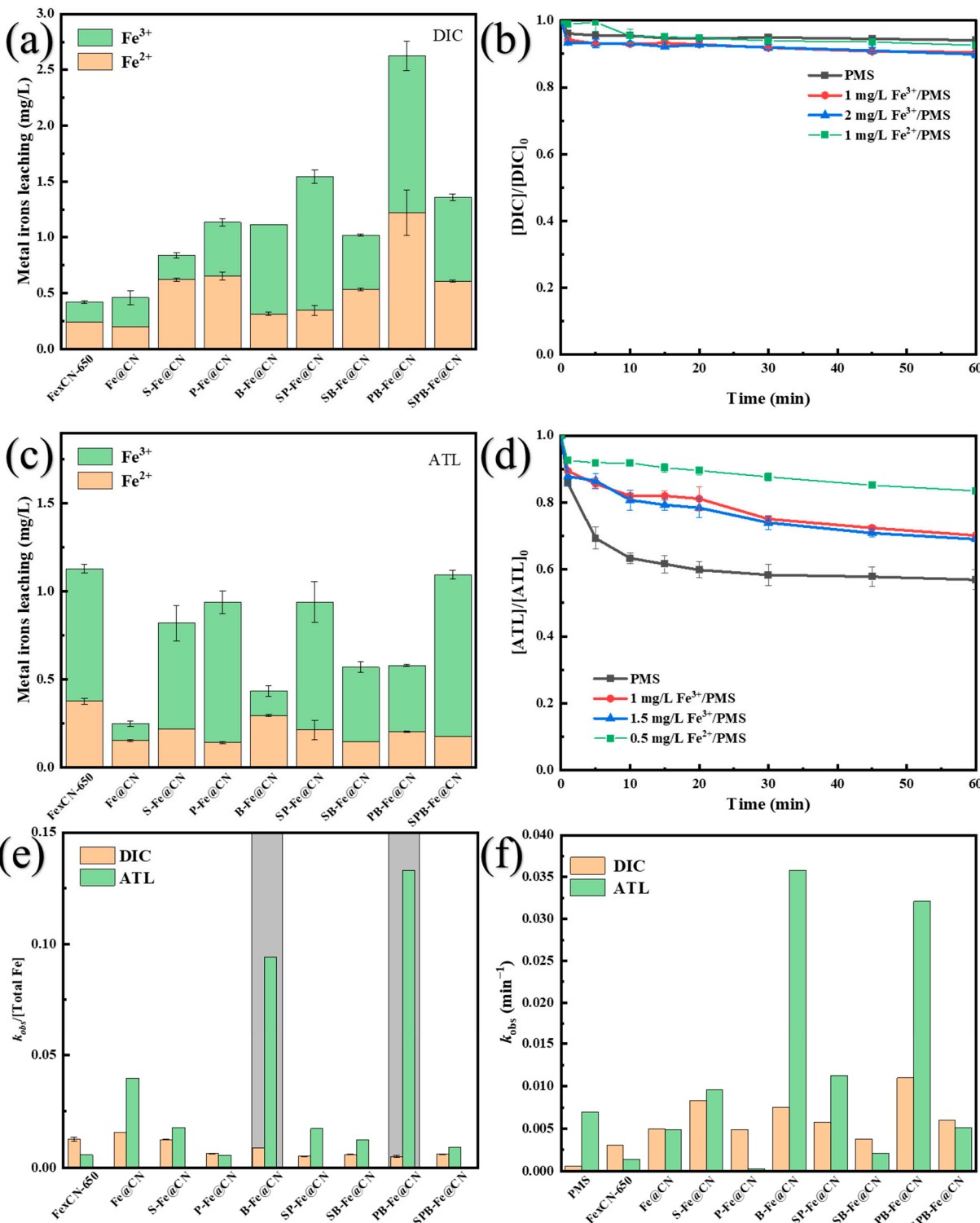

**Figure 7.** $Fe^{3+}$ and $Fe^{2+}$ leaching from (**a**) X-Fe@CN/PMS/DIC and (**c**) X-Fe@CN/PMS/ATL system; DIC (**b**) and ATL (**d**) removal in PMS activation by leached $Fe^{2+}$ or $Fe^{3+}$; (**e**) effect of X-Fe@CN on $k_{obs}$/(Total Fe) of DIC and ATL removal; (**f**) effect of X-Fe@CN (excluded iron ions) on $k_{obs}$ of DIC and ATL removal. Experimental conditions: $(Fe^{3+})$ = 1.0/1.5/2.0 mg/L or $(Fe^{2+})$ = 0.5/1.0 mg/L, (PMS) = 1.30 mmol/L, (DIC) = 8.84 mg/L, (ATL) = 10.65 mg/L, (X-Fe@CN) = 0.1 g/L, pH = 7.0.

To investigate the impact of dissolved iron ions on the performance of DIC removal by X-Fe@CN activating PMS, 1.0 mg/L of $Fe^{2+}$ and 1.0 mg/L and 2.0 mg/L of $Fe^{3+}$ were introduced into the reaction, because the leached concentration is similar. From Figure 7d, it is evident that the activation of PMS was less improved by 0.5 mg/L $Fe^{2+}$ and 1.0 mg/L and 2 mg/L $Fe^{3+}$, which increased by 1.54%, 3.55%, and 4.30%, respectively. It shows that the removal of DIC by X-Fe@CN-activated PMS is basically a heterogeneous reaction due to the leached iron ions showing no significant performance on PMS activation.

To investigate the role of dissolved iron ions in X-Fe@CN/PMS/ATL reaction, we added 0.5 mg/L $Fe^{2+}$ and 1.0 mg/L (and 1.5 mg/L) $Fe^{3+}$ to the reaction. The results, depicted in Figure 7d, demonstrate that the removal efficiency of ATL is reduced compared to that of PMS alone. Specifically, the inclusion of 0.5 mg/L of $Fe^{2+}$ and 1.0 mg/L and 1.5 mg/L of $Fe^{3+}$ resulted in performance reductions of 26.56%, 13.23% and 12.09%, respectively. The addition of iron ions inhibited the activation of ATL by PMS.

In the unadjusted solution pH experiment, PMS alone exhibited poor ATL removal. After solution pH adjustment to 7.0 by adding NaOH (1.0 mol/L), electrostatic attraction between ATL and PMS [42] allowed PMS removal of 43% of ATL. Alternatively, the addition of NaOH may have an alkaline activating effect on PMS. After adding iron ions, $H^+$ were produced by hydrolysis of $Fe^{2+}$ and $Fe^{3+}$ (Equations (2) and (3)). Decreasing the pH of the solution resulted in weakened PMS decomposition and thus weakened the electrostatic attraction between ATL and PMS, further inhibiting ATL removal. However, the removal efficiency of ATL increased with increasing $Fe^{3+}$ concentration, suggesting a certain activating effect of $Fe^{3+}$. Iron ions can participate in the reaction as Lewis acids (Equation (5)), which, in combination with electron-rich nucleophilic reagents (·R) (Equation (4)), facilitate the formation of reaction intermediates [12].

$$Fe^{3+} + 3H_2O \rightarrow Fe(OH)_3 + 3H^+, \tag{3}$$

$$Fe^{2+} + 2H_2O \rightarrow Fe(OH)_2 + 2H^+, \tag{4}$$

$$\cdot OH + RH \rightarrow \cdot R + H_2O, \tag{5}$$

$$Fe^{3+} + R \rightarrow Fe^{2+} + R^+. \tag{6}$$

The activation of PMS for ATL removal coexisted in both homogeneous and heterogeneous reactions in the presence of X-Fe@CN. When ATL was removed in the solution, the overall dissolution of iron ions was relatively lesser and the relative $k_{obs}$ values were higher, probably because the activation of PMS by metal ions in the homogeneous reactions reduced the performance of the reaction of $Fe^0$ and $Fe^{2+}$ with PMS in X-Fe@CN. The relative effects were better for B-Fe@CN and PB-Fe@CN. The efficacy and safety of X-Fe@CN in decomposing DIC and ATL were thoroughly evaluated by normalizing the rate constant ($k_{obs}$) with the concentration of dissolved iron ions. As shown in Figure 7e, $k_{obs}$/(total Fe) values of PB-Fe@CN and B-Fe@CN were the highest after 60 min of reaction, indicating that they have excellent removal performance at relatively low amounts of total dissolved Fe ions.

Rate constant $k_{obs}$ of the heterogeneous reaction was determined by eliminating the influence of activated PMS in the homogeneous reaction (Figure 7f). Upon removal of the ionic component, it became evident that PB-Fe@CN demonstrates the most outstanding DIC removal performance, which accomplishes the goal of enhancing Fe@CN's catalytic activity through PB heteroatom doping. For the elimination of ATL, both B-Fe@CN and PB-Fe@CN demonstrated outstanding catalytic efficiency, achieving the goal of enhancing the catalytic performance of Fe@CN through B/PB heteroatom doping. It can be seen that X-Fe@CN may have different removal effects on different pollutants, but PB-Fe@CN showed significant removal effects on both DIC and ATL. However, when considering the relative $k_{obs}$ under specific iron ion dissolution conditions, B-Fe@CN demonstrated superior catalytic activity and lower iron ion dissolution. In addition, XRD analysis showed that the

Fe$_2$O$_3$ and Fe$_3$O$_4$ contents of B-Fe@CN were significantly lower compared to Fe@CN, but retained a higher content of Fe$^0$. This led to a noticeable reduction in the dissolution of iron ions, resulting in a safer catalyst.

### 3.4. Effects of Initial Reaction Factors

In combination with the leaching of iron ions, B-Fe@CN and PB-Fe@CN were selected for the optimization of all reaction parameters. As shown in Figure 8, the removal efficiency of DIC exhibited an initial increase followed by a decrease with increasing the dosage of both DIC and ATL. The apparent reaction rate constant, $k_{obs}$, of DIC increased from 0.010 min$^{-1}$ to 0.021 min$^{-1}$ when the dosage of B-Fe@CN was increased from 100 mg/L to 200 mg/L. Similarly, an increase in dosage of PB-Fe@CN from 100 mg/L to 200 mg/L resulted in an increase in $k_{obs}$ of DIC from 0.010 min$^{-1}$ to 0.021 min$^{-1}$. The apparent reaction rate constant $k_{obs}$ for DIC increased from 0.013 to 0.017 min$^{-1}$ with an increase in PB-Fe@CN dosage from 100 to 200 mg/L. The most favorable outcomes were attained utilizing 200 mg/L. The apparent reaction rate constant $k_{obs}$ for DIC rose from 0.010 to 0.021 min$^{-1}$ with PB-Fe@CN of 200 mg/L. The ideal efficiency was attained using 200 mg/L B-Fe@CN, which managed to disintegrate 74.8% of DIC. With an increase in catalyst dosage, the number of active catalytic sites also increased [45,46]. This, in turn, increased the contact area between the activator and both DIC and PMS, which contributed towards the acceleration of PMS decomposition and the production of significantly larger amounts of reactive species. As a result, the removal of DIC was accelerated. However, the apparent reaction rate constant $k_{obs}$ of DIC decreased at the catalyst dosage of 300 mg/L and 400 mg/L. The increase in catalyst dosage was found to be the cause of the decrease in the apparent reaction rate constant $k_{obs}$ of DIC. It was determined that increasing the catalyst dosage excessively could result in the generation of a large amount of sulfate radicals in the solution. Additionally, the quenching of sulfate radicals by PMS and the self-quenching reaction between sulfate species (as shown in Equations (7) and (8)) could also lead to the formation of a significant amount of sulfate radicals.

$$HSO_5^- + SO_4^{\cdot-} \rightarrow SO_5^{\cdot-} + SO_4^{2-} + H^+, \tag{7}$$

$$SO_4^{\cdot-} + SO_4^{\cdot-} \rightarrow S_2O_8^{2-}. \tag{8}$$

For the removal of ATL, the dosage of B-Fe@CN at 100 mg/L showed the highest apparent reaction rate constant with the $k_{obs}$ of 0.015 min$^{-1}$. Upon increasing the dosage of B-Fe@CN to 200 mg/L, the $k_{obs}$ of ATL decreased rapidly to 0.05 min$^{-1}$, and a further increase in dosage did not affect $k_{obs}$. Increasing the dosage of PB-Fe@CN from 200 mg/L to 400 mg/L resulted in a rapid increase in the apparent reaction rate constant $k_{obs}$ of ATL from 0.015 min$^{-1}$ to 0.044 min$^{-1}$. Additionally, the removal rate of ATL increased from 71.9% to 96.0%. Continued increases in the dosage of PB-Fe@CN had an inhibitory effect. According to Figure 8e, the main reason for this effect may be the limited concentration of PMS, which only accelerated the rate of production of free radicals with the continued increase in catalyst dosage. Better removal was achieved after 30 min; however, at 60 min, removal efficiency was slightly reduced due to the inability to generate additional free radicals or quench free radicals. PB-Fe@CN was found to remove ATL much more effectively than B-Fe@CN when both activators were dosed optimally. PB-Fe@CN demonstrated significantly better removal of ATL than B-Fe@CN at the optimal dosage (0.044 min$^{-1}$ > 0.015 min$^{-1}$).

The removal efficiencies of DIC and ATL were significantly increased as PMS dosing increased, as depicted in Figure 8g–i. In the absence of PMS, the adsorption efficiency of B-Fe@CN on DIC was 10.88% and changed to 18.41% for PB-Fe@CN adsorption ATL. The $k_{obs}$ value for DIC removal increased from 0.0044 min$^{-1}$ to 0.0207 min$^{-1}$, while that for ATL removal rose from 0.0059 min$^{-1}$ to 0.0435 min$^{-1}$ with the PMS dosage increasing from 0.65 mM to 1.30 mM. The optimal dosage of PMS was selected as 1.30 mM, given that the removal efficiency for ATL was 0.0435 min$^{-1}$. The improved performance can be attributed to the release of sufficient reactive oxygen species (ROS) by PMS [47], which

promotes the removal of DIC and ATL. PMS (1.30 mM)/B-Fe@CN (200 mg/L)/DIC and PMS (1.30 mM)/PB-Fe@CN (400 mg/L)/ATL were chosen for subsequent studies.

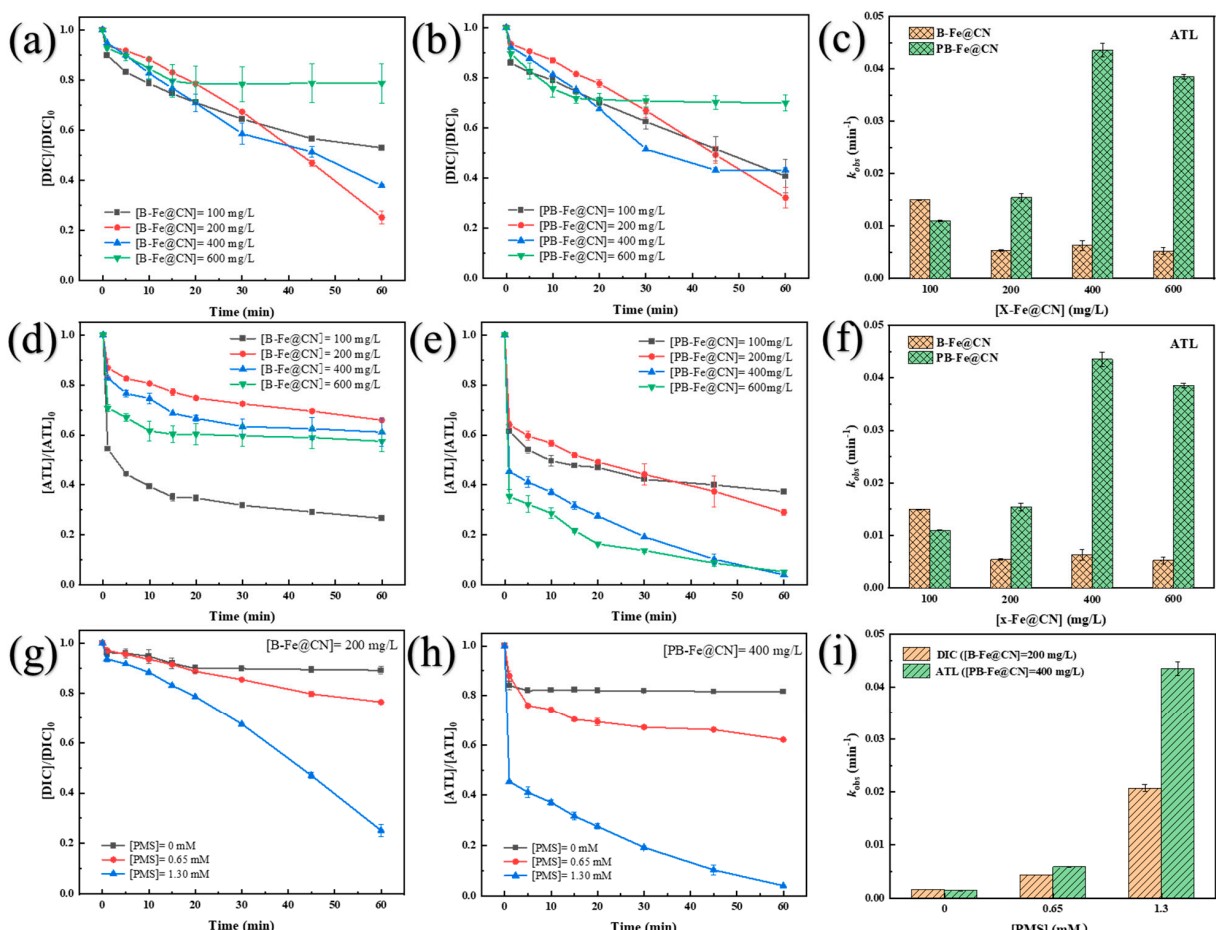

**Figure 8.** Effect of B-Fe@CN and PB-Fe@CN dosing on DIC and ATL removal (**a–f**); Effect of PMS concentration on DIC and ATL removal (**g–i**). Experimental conditions: (DIC) = 8.84 mg/L, (ATL) = 10.65 mg/L, pH = 7.0.

### 3.5. Proposed Reaction Mechanism Due to Heteroatom Doping

#### 3.5.1. Identification of Surface-Active Sites

Doping heteroatoms is an effective method to provide additional defects and active sites in a PMS activator [16]. The characteristic Raman peaks of the graphene carbon material appear in the sample at 1330 cm$^{-1}$ and 1585 cm$^{-1}$, where the peak at 1330 cm$^{-1}$ corresponds to the $sp^3$ hybridization of the disordered structure and is named the D peak, and the peak at 1585 cm$^{-1}$ corresponds to the $sp^2$ hybridization of the ordered structure and is named the G peak [48]. The ratio of intensity of D and G peaks ($I_D$:$I_G$) is used to indicate the degree of defects in the carbon material, with a larger ratio indicating a greater degree of defects in the carbon. As shown in Figure 9a, the $I_D$:$I_G$ value of Fe@CN compared to FexCN-650 increased from 1.17 to 1.23. This indicates that the coupling of carbon nitride results in the presence of more defects and a disordered structure in the catalyst after g-C$_3$N$_4$ introduction. The $I_D$:$I_G$ values of B-Fe@CN and PB-Fe@CN increased to 1.27 and 1.28, while for P-Fe@CN compared to Fe@CN the value decreased to 1.18. Co-doping with both P and B along with N created more defects, while co-doping with P and N resulted in fewer defects compared to N doping alone. This outcome is in line with Duan et al.'s research on graphene doped with heteroatoms [17]. It is worth noting that Figure 10a demonstrates a direct correlation between $k_{obs}$ and $I_D$:$I_G$ for DIC ($R^2$ = 0.86) and ATL ($R^2$ = 0.94) removal. This suggests that the greater the defect rate of X-Fe@CN,

the better its catalytic performance. The defects in the catalyst can disturb the inertness of the $sp^2$ lattice and serve as locations for accommodating adsorption and activating PMS [17]. Additionally, doping with B and PB further disturbs the $sp^2$-hybridized carbon conformation in the graphene-like structure. The radii of P and B atoms are larger than those of C and N, leading to more local structural distortions in the carbon material and increasing the number of defects. In contrast, P doping results in greater graphitization and fewer defects in the material.

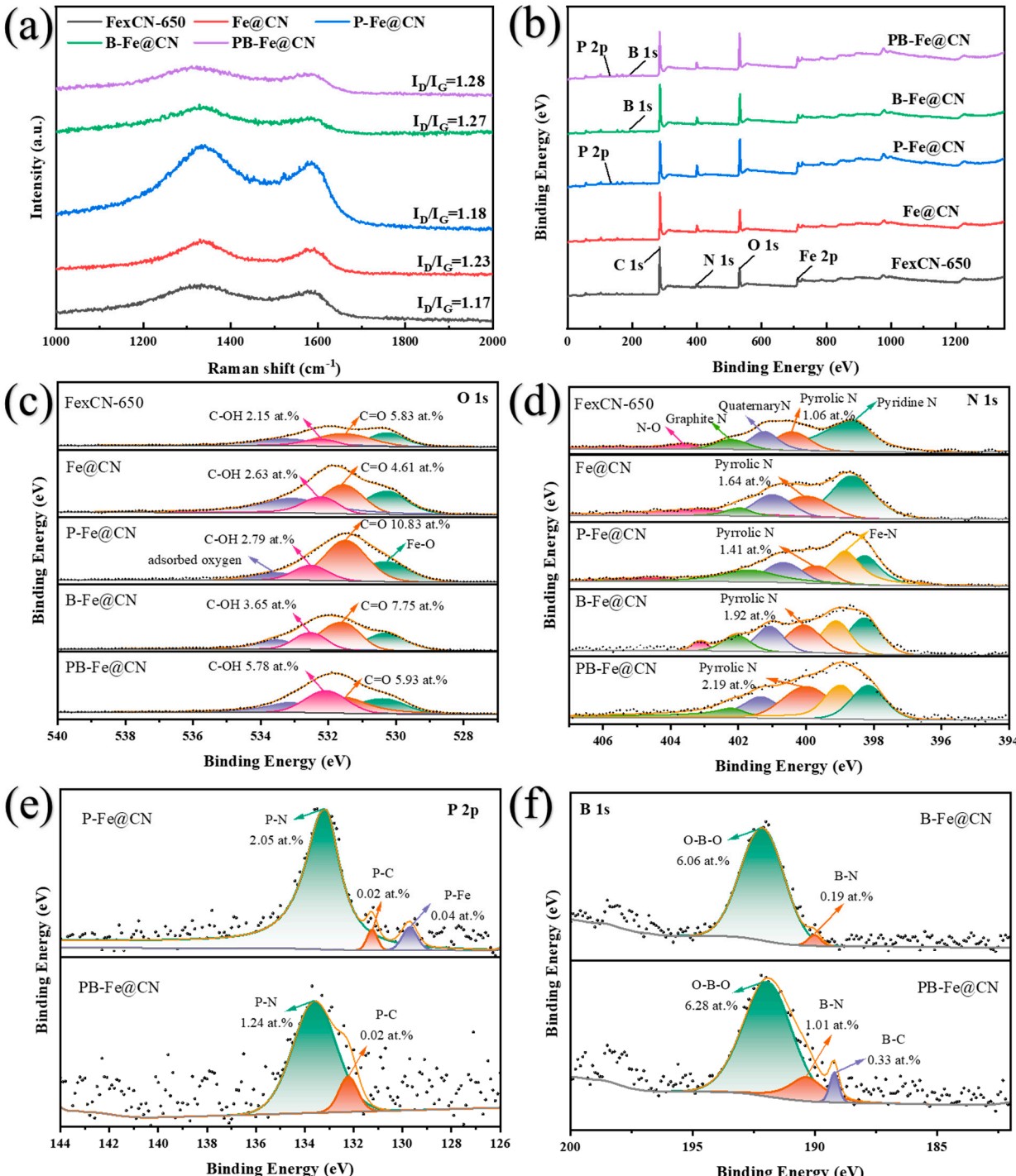

**Figure 9.** (**a**) Raman and XPS (**b**) survey, (**c**) O 1s, (**d**) N 1s spectra of FexCN-650, Fe@CN, P-Fe@CN, B-Fe@CN and PB-Fe@CN; (**e**) XPS P 2p spectra of P-Fe@CN and PB-Fe@CN; (**f**) XPS B 1s spectra of B-Fe@CN and PB-Fe@CN.

Four major peaks corresponding to C 1s (284.0 eV), N 1s (400.1 eV), O 1s (531.7 eV), and Fe 2p (712.0 eV) are observed in Figure 8b [12]. Weaker peaks of P 2p (135.0 eV) and B 1s (193.0 eV) also appear, indicating successful doping of the Fe@CN material with heteroatoms P and B, respectively. The atomic contents of C, N, O, Fe, P, B for all fabricated catalysts obtained from Figure 8b are listed in Table S6. Figure S1 shows the XPS spectra of C 1s in FexCN-650, Fe@CN, P-Fe@CN, B-Fe@CN, and PB-Fe@CN. The C 1s peak located at 284.7 eV corresponds to the C-C bond of $sp^2$ hybridized graphitic carbon. The C-C bond decreases after heteroatom doping, which is related to the substitution of C atoms by heteroatoms (Table S7). The figure displayed in Figure S2 presents the spectrum of Fe 2p; Fe can be seen at the peak of 707 eV. Additionally, Fe $2p_{3/2}$ and Fe $2p_{1/2}$ are denoted by the peaks at 711.3 and 722.7 eV, respectively. Further analysis reveals that the peaks of Fe $2p_{3/2}$ deconvolute to two peaks at 711.2 and 709.2 eV, and the peaks of Fe $2p_{1/2}$ deconvolute to 724.8 and 722.8 eV, which is evidence of the presence of $Fe^{2+}$ and $Fe^{3+}$ in $Fe_3O_4$. After co-doping, the zero-valent Fe content considerably increased, potentially serving as a site for PMS activation. However, the content of $Fe^0$ in P-Fe@CN was enhanced, while its catalytic activity decreased (Table S8). It can be concluded from Figure S3 that there was no correlation between the content of $Fe^0$ and the $k_{obs}$ removed by DIC and ATL. The reason may be that $Fe^0$ is mainly encapsulated within the framework structure, and it is difficult to contact with the PMS to play the role of activating the PMS. It has been demonstrated that due to the small pore size (3–8 nm) of the carbon shell, the internal $Fe^0$ in the catalyst has no access to PMS and contaminants, thus making it difficult to function [49].

Surface oxygen plays a vital role in activating the persulfate [50]. Figure 9c shows the XPS spectra for the O 1s of all fabricated novel materials. The peak at 530.1 eV represents the Fe-O bond in $Fe_2O_3$ and $Fe_3O_4$ (the lattice oxygen). The peak at 532.8 eV identifies the C=O of the catalysts, while the peak at 533.7 eV represents the surface hydroxyl group, C-OH or the surface hydroxyl group of iron oxides. The addition of different atoms to the catalyst uncovered more hydroxyl groups on the surface (Table S9), and the quantity of -OH groups was closely linked to the breakdown of DIC, as presented in Figure 10b ($R^2$ = 0.97). Prior findings suggest that -OH groups can become organic radicals by relinquishing protons, which then triggers the formation of $SO_4^{\bullet-}$ (Equations (3) and (4)) [31]. Therefore, we surmise that -OH groups might help stimulate the generation of free radicals by PMS.

$$\text{surface} - \text{OH} + S_2O_8^{2-} \rightarrow SO_4^{\cdot-} + \text{surface} - O^{\cdot} + HSO_4^{-}. \tag{9}$$

The XPS fine spectrum of N 1s is presented in Figure 9d, including pyridine nitrogen at 398.6 eV, Fe-N at 399.3 eV, pyrrole nitrogen at 400.0 eV, graphitic nitrogen at 402.5 eV, quaternary ammonium nitrogen at 401.1 eV, and the N-O bond at 403.5 eV [12,48,49]. Nitrogen atoms can be introduced into the carbon ring during pyrolysis, leading to the rearrangement of the carbon structure. Heteroatom doping increases the number of Fe-N sites, which act as metal ion centers for the initial adsorption of oxygen molecules and PMS and as reaction intermediates [7] (Table S10). It has also been reported that the radical pathway may occur at defective sites of carbon catalysts, Lewis base sites such as pyrrolic/pyridine N [18]. The contents of the different pyrrolic Ns in FexCN-650, Fe@CN, P-Fe@CN, B-Fe@CN, and PB-Fe@CN were found to be 1.06 at.%, 1.84 at.%, 1.41 at.%, 1.92 at.%, and 2.19 at.% at.%, respectively (Figure 9d). The content of pyridine nitrogen showed a significant positive correlation with the catalyst's activity ($R^2_{DIC}$ = 0.92, $R^2_{ATL}$ = 0.74) (Figure 10c), which suggests that pyrrolic N is an important active site for the activation of PMS. However, the contents of pyridine N, Fe-N, quaternary N, graphite N, and N-O in the catalysts did not show a significant relationship with the catalytic activity.

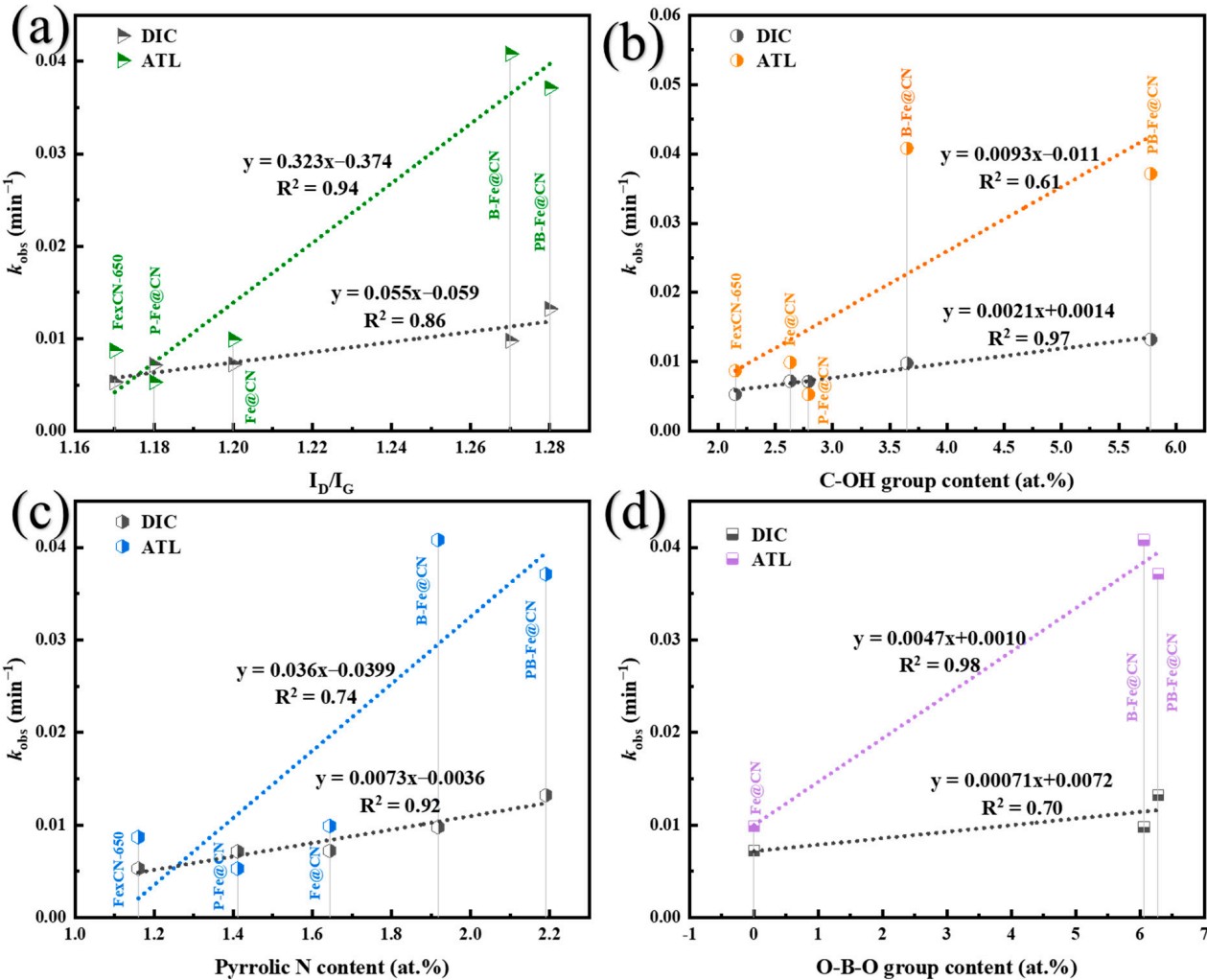

**Figure 10.** The correlation of $k_{obs}$ of DIC or ATL removal to (**a**) $I_D/I_G$, (**b**) C-OH bond, (**c**) Pyrrolic N, (**d**) O-B-O bond content.

As shown in Figure 9e, the peak at 133.2 eV corresponds to P-N, the peak at 132.4 eV corresponds to P-C, and the peak at 129.4 eV corresponds to Fe-P [20,39]. XPS results indicate that P atoms replace C atoms. The appearance of P-N suggests that P may have replaced C and successfully doped into the C backbone during thermal polymerization [51]. The P-C bond also confirms the incorporation of P elements in the carbon matrix [39]. It has been shown that Fe-P is the major active site after phosphorus doping [52], but P-Fe@CN did not show any enhancement in catalytic performance. The content of P-N in P-Fe@CN and PB-Fe@CN was dominant, but the content of P-N did not correlate with the $k_{obs}$ of the catalysts for the removal of DIC and ATL, as can be seen in Figure S4. There are two possible reasons for this. On the one hand, it is more difficult to introduce P into a carbon structure than to introduce N because of its relatively large size and weaker ability to attract electron [51]. As a result, the catalysts prepared by doping phosphorus in the carbon skeleton had low content, making it difficult to activate them. This, in turn, led to the lack of improvement of catalytic activity in P-Fe@CN. On the other hand, it was also shown in the current study that P-doped carbon catalysts may be relatively unresponsive to PMS activity, with P groups effectively scavenging $SO_4^{\bullet-}$ and ·OH, leading to difficulties in degrading pollutants [16].

As shown in Figure 9f, the peak at 192.3 eV corresponds to O-B-O, the peak at 190.0 eV corresponds to B-N, and the peak at 189.3 eV corresponds to B-C [22,47]. The bond of O-B-O can serve as both an adsorption and an activation site for PMS by interacting with the N portion to enhance the overall electronic properties of the PMS-activated carbon structure [53]. The presence of B-C bonds indicates successful doping of B into the carbon and nitrogen backbones. The content of O-B-O in B-Fe@CN and PB-Fe@CN was predominant, and as shown in Figure 10d, the content of O-B-O was significantly correlated with the $k_{obs}$ of catalysts for ATL removal ($R^2 = 0.98$). PB-Fe@CN has a higher overall boron content (7.32 at.% > 6.24 at.%) compared to B-Fe@CN and contains more O-B-O, B-N and B-C, thus demonstrating a superior catalytic effect in the removal of ATL [19,22]. O-B-O can act as a Lewis acid group to promote activation and adsorption of PMS, which is an important influencing factor for the significant catalytic effect of both catalysts [22,31].

3.5.2. Identification of Reactive Oxygen Species

Electron spin resonance (ESR) experiments were conducted on B-Fe@CN/PMS and PB-Fe@CN/PMS, respectively (Figure 10). B-Fe@CN and PB-Fe@CN were capable of activating PMS, causing the production of $SO_4^{\bullet-}$ and $\cdot OH$. As shown in Figure 10a, at the fifteen-minute mark, DMPO-OH reached maximum intensity, after which the signal strength started to decrease, indicating that $\cdot OH$ was being utilized during the reaction. The inclusion of PMS lead to a decrease in solution pH, which resulted in a slower conversion of $SO_4^{\bullet-}$ to $\cdot OH$ and a subsequent reduction in signal intensity of $\cdot OH$ (Equation (10)) [34]. The signal intensity of DMPO-$SO_4^{\bullet-}$ remained relatively stable after 15 min, indicating continuous $SO_4^{\bullet-}$ production during the reaction. PB-Fe@CN activation of PMS to produce $SO_4^{\bullet-}$ and $\cdot OH$ is similar to that of B-Fe@CN, but the total process is faster, with the highest point of pre-production of $\cdot OH$ at 5 min.

$$SO_4^{\cdot-} + OH^- \rightarrow SO_4^{2-} + \cdot OH, \tag{10}$$

$$HSO_5^- + SO_5^{2-} \rightarrow HSO_4^- + SO_4^{2-} + {}^1O_2. \tag{11}$$

According to Figures 10b and 11d, when PMS is present alone, the TEMPO's characteristic signal with three peaks and a peak intensity of 1:1:1 is visible. This indicates that ${}^1O_2$ generation occurs due to the self-decomposition of PMS (Equation (11)). Following the addition of PB-Fe@CN, the signal intensity of ${}^1O_2$ diminished gradually until it disappeared, thus suggesting a possible lack of ${}^1O_2$ generation in this PMS activation process. After the addition of B-Fe@CN, the signal intensity of ${}^1O_2$ reaches its maximum at 5 min and then decreases rapidly. The possible reason for the rapid decrease in ${}^1O_2$ after 5 min is that water molecules tend to consume or quench ${}^1O_2$ during the reaction [54]. Due to $O_2^{\bullet-}$ disproportionation, obtaining the DMPO-$O_2^{\bullet-}$ signal in an aqueous system was difficult; thus, ethanol was used as the reaction medium instead of water. No clear ESR signal of $O_2^{\bullet-}$ was found in both catalyst-activated PMSes under PMS/DMPO conditions, as shown in Figure S5. It is worth noting that $SO_4^{\bullet-}$ and $\cdot OH$ are usually produced in a series of reactions involving electron transfer from iron (Equations (12)–(16)) [32,55–57].

$$2HSO_5^- + Fe^0 \rightarrow Fe^{2+} + 2SO_4^{\cdot-} + 2OH^-, \tag{12}$$

$$Fe^{2+} + HSO_5^- \rightarrow Fe^{3+} + OH^- + SO_4^{\cdot-}, \tag{13}$$

$$HSO_5^- + Fe^{3+} \rightarrow Fe^{2+} + SO_5^{\cdot-} + H^+, \tag{14}$$

$$Fe^0 + O_2 + 2H^+ \rightarrow Fe^{2+} + H_2O_2, \tag{15}$$

$$Fe^{2+} + H_2O_2 \rightarrow Fe^{3+} + OH^- + \cdot OH \tag{16}$$

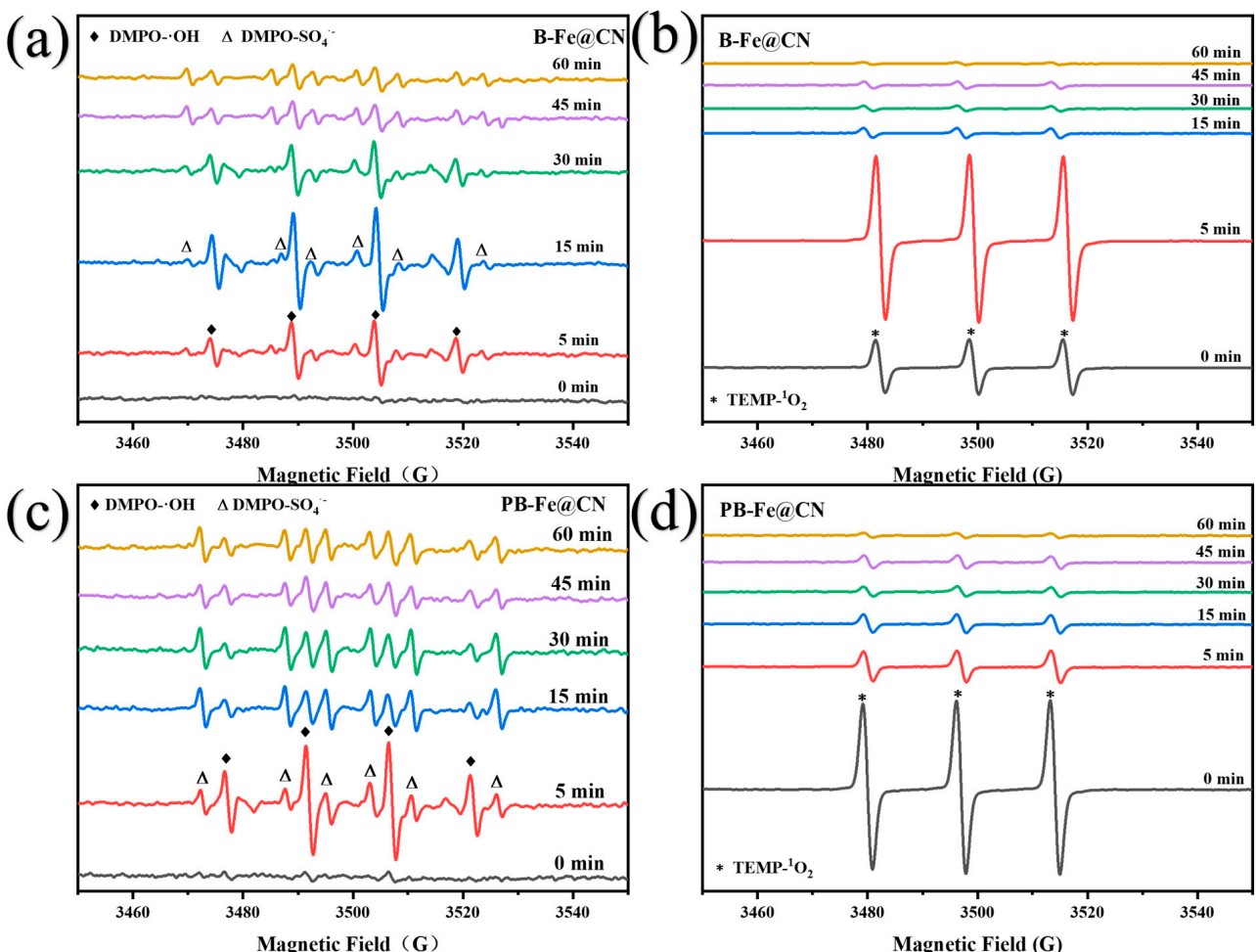

**Figure 11.** EPR spectra obtained using (**a**) DMPO and (**b**) TEMP as spin-trapping agents in B-Fe@CN/PMS; EPR spectra obtained using (**c**) DMPO and (**d**) TEMP as spin-trapping agents in PB-Fe@CN/PMS. Experimental conditions: (PMS) = 1.30 mM, (B-Fe@CN) = 200 mg/L, (PB-Fe@CN) = 400 mg/L, initial pH = 7.0.

3.5.3. Identification of Electronic Transfers

Electrochemical studies such as CV, LSV, Tafel and EIS were also used to confirm the existence of electron transfer mechanisms (Figure 11). Surface electron transfer is a representative non-radical pathway in PMS activation by carbon materials, in which carbon materials act as electron carriers to transfer electrons from electron-rich contaminant (electron donors) to PMS (electron acceptors). To further evaluate the relationship between electron transfer performance and catalytic performance, the response current values and charge transfer resistances of various catalysts were determined. Figure 12a shows the CV curves of FexCN-650, Fe@CN, P-Fe@CN, B-Fe@CN and PB-Fe@CN, in which the order of current intensity is PB-Fe@CN > Fe@CN > FexCN-650 > P-Fe@CN > B-Fe@CN under the same voltage condition. The results showed that PB-Fe@CN has a larger response current value than the other catalysts, which is favorable to the electron transfer on the electrode surface and thus can accelerate the removal of the contaminant [58]. In contrast, the current intensity of B-Fe@CN is lower than that of Fe@CN, which has a reduced electron transfer capacity.

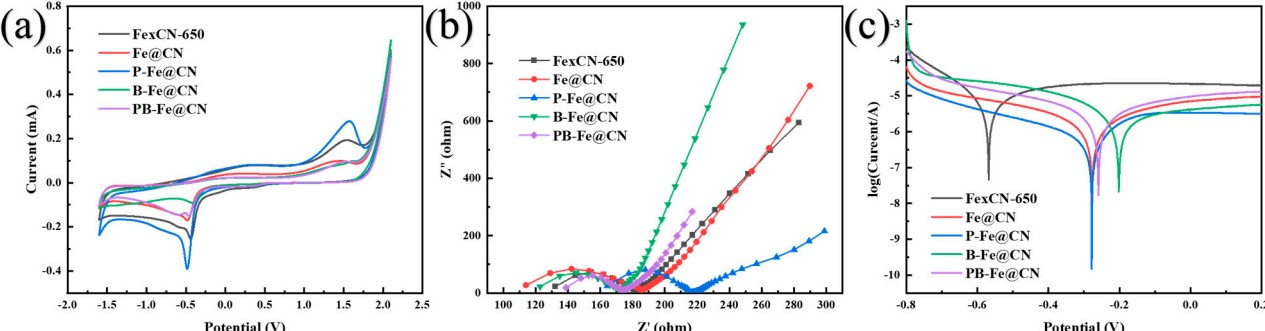

**Figure 12.** (**a**) The cyclic voltammetry curve, (**b**) the EIS curve, (**c**) the Tafel curves of FexCN-650, Fe@CN, P-Fe@CN, B-Fe@CN and PB-Fe@CN.

The charge transfer resistances of various catalysts were evaluated (Figure 12b). The electrochemical impedance (Rct) declined in the following order: Fe@CN > B-FexCN > P-Fe@CN > Fe@CN-650 > PB-Fe@CN (Table 2). The material exhibiting a smaller electrochemical impedance value is more conducive to electron transfer and facilitates improved contaminant removal in PMS activation [31]. Compared to FexCN-650, the inclusion of both P and B decreased the catalyst's impedance. Additionally, the charge transfer resistance significantly decreased with PB co-doping, indicating the synergistic influence of P and B in enhancing electron transport in the catalyst.

**Table 2.** Rs and Rct values for FexCN-650, Fe@CN, P-Fe@CN, B-Fe@CN and PB-Fe@CN.

| Catalyst | Rs ($\Omega$ cm$^{-2}$) | Rct ($\Omega$ cm$^{-2}$) |
|---|---|---|
| FexCN-650 | 51.13 | 127.6 |
| Fe@CN | 39.29 | 134.3 |
| P-Fe@CN | 43.32 | 128.7 |
| B-Fe@CN | 38.55 | 130.3 |
| PB-Fe@CN | 82.01 | 79.52 |

The Tafel curve can be used to evaluate the corrosion susceptibility of the anode [59]. It is commonly accepted that the sample's corrosion resistance is better if its self-corrosion potential is more positive and its self-corrosion current density is smaller [60]. The result of Figure 12c revealed notable variations among X-Fe@CN anodes with distinct catalyst loadings. FexCN-650 exhibits the least corrosion potential, whereas Fe@CN displays a considerably higher corrosion potential. These findings suggested that the cohesion of carbon nitride promotes the catalyst's stability. The corrosion potentials of P-Fe@CN ($-0.28$ V) and Fe@CN ($-0.27$ V) are similar, but PB-Fe@CN ($-0.26$ V) and B-Fe@CN ($-0.20$ V) loaded anodes show higher positive corrosion potentials. These findings indicated that the PB-Fe@CN and B-Fe@CN loaded electrodes exhibit greater corrosion resistance. Moreover, the results proved that doping Fe@CN with the B element is advantageous for enhancing corrosion resistance, whereas doping with the P element is not. B-Fe@CN exhibits higher corrosion resistance than PB-Fe@CN, thereby confirming that the addition of phosphorus decreases its corrosion resistance.

Above all, compared to the other three prepared catalysts, B-Fe@CN exhibits higher resistance and minimal current response values, suggesting that its surface electron transfer capacity is poor and its non-radical pathway is mainly through the action of $^1O_2$ acting. The doping of B introduces more defects, providing O-B-O and pyrrolic nitrogen active sites for adsorption of PMS and contaminants and generating free radicals and $^1O_2$ to further remove the contaminants. In contrast, PB-Fe@CN shows the smallest resistance value and the most significant current response value and catalytic performance. The enhanced electron mobility on the graphitized carbon structure can be attributed to moderate amounts of P and B co-doping. This results in the breakage of the surface inertia of the carbon

material and the lack of electrons around the P and B heteroatoms, providing P doping in the carbon structure [19]. The introduction of phosphorus decreases the impedance of the catalyst, while the addition of boron enhances its corrosion resistance. Upon B doping, the increased defects and surface-active sites acted like a switch to initiate the response to PMS activation. Further doping with P resulted in an increase in electron transfer in the catalyst, which further improved the ability of the catalyst to activate PMS. The co-doping of PB created a remarkable synergistic effect, resulting in outstanding electron transfer capability and stability of PB-Fe@CN.

## 4. Conclusions

In summary, we successfully synthesized multi-heteroatom doped Fe@CN by coupling heteroatom-doped (S, P, B) carbon nitride and $NH_2$-MIL-53(Fe) via hydrothermal reaction followed by pyrolysis. A stable spindle morphology and a higher content of zero-valent iron were deemed necessary for the performance development of PMS activation.

However, P doping alone, S and B co-doping resulted in disruptions to spindle morphology of the proposed X-Fe@CN, whereas S doping decreased the content of $Fe^0$. B doping alone, P and B doping retained better spindle morphology and produced more defects and disordered structures. After optimization of the fabrication conditions, B-Fe@CN (200 mg/L) removed 74.8% of DIC, and PB-Fe@CN (400 mg/L) was able to remove 96% of ATL in 60 min under 1.30 mmol PMS dosing. X-Fe@CN removed DIC with ferrous ion dissolution below 3.0 mg/L and ATL below 1.5 mg/L, and B-Fe@CN and PB-Fe@CN showed higher $k_{obs}$/(Total $Fe^{3+}$). B doping alone and PB co-doping provided better optimization of the catalysts relative to other heteroatom doping methods, but there were also some differences in removal performance and iron ion leaching. B-Fe@CN had less relative iron ion solubilization and was safer, but less effective in degrading ATL. PB-Fe@CN had better removal performance for both pollutants. Therefore, we further investigated the effects and differences of P and B doping on the possible activation mechanisms of the catalysts.

We saw that B doping first provided the catalyst with a large number of active sites and defects such as O-B-O, C-OH, pyrrolic nitrogen, etc., producing and improving the corrosion resistance of the catalyst. These active sites successfully activated PMS, producing both $SO_4^{\bullet -}$, $\cdot OH$ and $^1O_2$. The introduction of P alone did not provide effective activity. However, after the introduction of B provided to the sites and defects to activate PMS, the introduction of P further enhanced the surface electron transfer ability of the catalyst, thus accelerating the process of activating PMS. Therefore, PB co-doping created a synergistic effect that enabled PMS activation toward the non-radical pathway and offered a higher level of corrosion resistance. The current investigation introduced a novel approach to fabricating iron-coupled graphitic carbon nitride catalysts doped with multi-heteroatoms. We optimized the fabrication and performance of the catalysts by screening for suitable heteroatom doping. The study promoted the understanding of the interactions between heteroatom doping and synergistic and antagonistic interactions and suggested new perspectives for improving catalyst performance through multi-heteroatom doping.

**Supplementary Materials:** The following supporting information can be downloaded at: https://www.mdpi.com/article/10.3390/w15244241/s1, Figure S1: The C 1s high-resolution XPS spectra of prepared (a) FexCN-650, (b) Fe@CN, (c) P-Fe@CN, (d) B-Fe@CN and (e) PB-Fe@CN; Figure S2: The Fe 2p high-resolution XPS spectra of prepared (a) FexCN-650, (b) Fe@CN, (c) P-Fe@CN, (d) B-Fe@CN and (e) PB-Fe@CN; Figure S3: The correlation of $k_{obs}$ of DIC and ATL removal to $Fe^0$ content; Figure S4: The correlation of kobs of DIC and ATL removal to P-N bond content; Figure S5: EPR spectra obtained using DMPO as spin-trapping agents in anhydrous ethanol; Table S1: Reagents and dosage for the preparation of $X-C_3N_4$; Table S2: The list of abbreviations; Table S3: Diffraction angles and layer spacings corresponding to the (002) crystal planes of $X-C_3N_4$; Table S4: The predicted content of S, P and B without considering the losses during the preparation process; Table S5: Elemental content of prepared X-Fe@CN by EDS analysis; Table S6: The element compositions of each element of prepared FexCN-650, Fe@CN, P-Fe@CN, B-Fe@CN and PB-Fe@CN; Table S7: The relative content of C 1s species of prepared FexCN-650, Fe@CN, P-Fe@CN, B-Fe@CN and PB-Fe@CN; Table S8: The

relative content of Fe 2p species of prepared FexCN-650, Fe@CN, P-Fe@CN, B-Fe@CN and PB-Fe@CN; Table S9: The relative content of O 1s species of prepared FexCN-650, Fe@CN, P-Fe@CN, B-Fe@CN and PB-Fe@CN; Table S10: The relative content of N 1s species of prepared FexCN-650, Fe@CN, P-Fe@CN, B-Fe@CN and PB-Fe@CN; Text S1: EPR and Electrochemical experimental procedures.

**Author Contributions:** Conceptualization, J.C.; data curation, J.C. and R.R.; formal analysis, J.C., R.R. and Y.L.; funding acquisition, C.L., Z.W. and F.Q.; investigation, J.C., R.R. and Y.L.; methodology, J.C., R.R., C.L. and Z.W.; project administration, F.Q.; resources, F.Q.; supervision, Y.L., C.L., Z.W. and F.Q.; validation, J.C.; writing—original draft, J.C.; writing—review and editing, R.R. and F.Q. All authors have read and agreed to the published version of the manuscript.

**Funding:** This research was funded by the National Key Research and Development Program of China (2021YFE0100800), the National Natural Science Foundation of China (Nos. 22076012, 52370064, 52200035, 52100002 and 52300001), the Beijing Forestry University Outstanding Young Talent Cultivation Project (No. 2019JQ03008), China Postdoctoral Science Foundation (2023T160053, 2023M730275 and 2021M700448), and the Yangtze River Joint Research Phase II Program (2022-LHYJ-02-0510-02).

**Data Availability Statement:** The authors confirm that the data supporting the findings of this study are available within the article (and its Supplementary Materials).

**Conflicts of Interest:** The authors declare no conflict of interest.

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
