# Peer review of "Multi-Heteroatom Doped Fe@CN Activation Peroxomonosulfate for the Removal of Trace Organic Contaminants from Water: Optimizing Fabrication and Performance"

_water, doi:10.3390/w15244241_

Round 1

Reviewer 1 Report

Comments and Suggestions for Authors

The manuscript entitled " Multi-heteroatom doped Fe@CN activation PMS for the degradation of trace organic contaminants from water: fabrication optimization and performance" shows important information about the material and the application. This article presents a new technological scheme of multi-heteroatom doped X-Fe@CN activated PMS for the effective treatment and elimination of TOrCs. The study exhibits the effect of heteroatom doping on the structure and performance of the catalyst. In addition, the results show the synergistic and antagonistic effects between the doped heteroatoms, especially providing new insights into the synergistic mechanism of P and B. In my opinion, there are various points that could be re-considered before publication in any case:

1. In the concluding part of the abstract, you discuss the implications and future direction of the research. It would be beneficial to elaborate on the broader implications of your findings, such as potential applications in wastewater treatment processes or other relevant fields.

2. In the process of discussing the potential solution to TOrCs, you have referred to transition metal nitrogen-carbon materials as an effective alternative. This appears to be where your study comes in. It would be beneficial to elaborate on why these materials are more suitable, and perhaps provide some examples where they have previously been utilized successfully.

3. In the Materials and Methods section, it is recommended that procedures for EPR and electrochemical experiments be provided in the Supplementary Materials.

4. Your manuscript needs careful editing and particular attention to English grammar, spelling, and sentence structure. There is at least one Spelling error in the manuscript, such as, in page 15, “Figure 9(i)” would be “Figure 9i”. In page 20, “1O2” would be “1O2”. In line 117, please change “was” to “were”. Please check the manuscript carefully. Consider clarifying the abbreviations used in the text, such as SR-AOPs (in page 1), etc.

Author Response

请参阅附件。

Reviewer 2 Report

Comments and Suggestions for Authors

In this manuscript, the authors synthesized various multi-heteroatom doped Fe@CN (X-Fe@CN) materials and assessed their efficacy in the photodegradation of dicamba (DIC) and atenolol (ATL) by activating PMS. The materials were thoroughly characterized, the investigation was meaningful and systematic, and the discussion was insightful.

However, a critical drawback in this manuscript is the lack of distinction between the effects of sorption removal and catalytic degradation. As indicated in line 132, X-Fe@CN and PMS were added simultaneously during the reaction, suggesting that the abrupt decline in C/C0 in Figure 5a & b might be attributed to sorption by the X-Fe@CN material. The coexistence of sorption and catalytic degradation makes it challenging to compare the catalytic performance of different X-Fe@CN materials. Figure 5c indicates that the authors used the overall performance (sorption + catalytic degradation) to calculate the apparent reaction rate, erroneously assigning sorption removal to catalytic degradation. This raises concerns about the performance comparison and overall discussion. The authors should separate sorption removal from catalytic degradation by adding the catalyst first, stirring in the dark to achieve sorption equilibrium, and then introducing PMS for the catalytic reaction. Please use only the catalytic degradation performance to calculate the reaction rate and compare different catalysts, and revise all relevant comparison and discussion sections (Figure 5-8 and the corresponding discussions). It requires significant revision.

Additionally, some experimental details were missing from the manuscript. In Section 2.2.1, the authors should provide the nominal weight or molar percent of S, P, and B in the doped materials. Furthermore, the actual percentage in the synthesized material, characterized by methods such as ICP, should be disclosed. Do these values align with the nominal percentages? Moreover, the equation for calculating the reaction rate constant is absent from both the manuscript and the supplementary information (SI). These missing details need to be included for a more comprehensive presentation. 

Reviewer 3 Report

Comments and Suggestions for Authors

In this manuscript the study of authors focuses on evaluating the effectiveness of Advanced Oxidation Processes (AOPs) using X-Fe@CN to degrade two types of Targeted Organic Contaminants (TOrCs), namely atenolol (ATL) and dicamba (DIC). DIC is a hydrophilic herbicide known for its easy solubility in water, which can result in its migration within agricultural ecosystems, leading to soil and groundwater contamination and posing risks to water safety and human health. ATL, on the other hand, is a β-blocker used in the treatment of cardiovascular illnesses, commonly found in various environmental samples, including wastewater, surface water, groundwater, soil, and sediments, with levels reaching up to 300 μg·L-.

The primary objective of the study is to achieve efficient degradation of these TOrCs. To accomplish this, the researchers aim to optimize the selection of the most suitable multi-heteroatom doped Fe@CN, fine-tune the fabrication parameters of optimized Fe@CN, and optimize the activation parameters of the peroxymonosulfate (PMS) reaction. The study also plans to analyze the surface-active sites and active species produced during the process and uncover the mechanism of TOrCs degradation by X-Fe@CN activating PMS.

Overall, the study intends to address the environmental concerns associated with DIC and ATL contamination by employing AOPs, specifically X-Fe@CN activating PMS. The optimization of various parameters and the investigation of degradation mechanisms are key aspects of the research. All the materials are well characterized, and the data shown seems very reliable. Hence it may be publishable in Water.

Reviewer 4 Report

Comments and Suggestions for Authors

The authors have synthesized a series of Fe@CN-based catalysts promoted by S, P and B elements for oxidation of trace organic contaminants in water using peroxomonosulphate as an oxidant. They have obtained a set of the catalysts modifided by all combinations of these heteroatoms followed by optimization of the process selecting a couple of the most efficient combinations. The article can be publushed after minor revision:

-the using abbreviations should be avoided in Abstract and the more - in article title. Not all abbreviations have been decrypted in the Introduction. It would be better to add the abbreviation list since the article is enriched with them;
- the authors affirm that boron introduces into carbon-containing phase to form B-C bonds, but it is widely known that B3+ is known to be very difficult to reduce https://doi.org/10.1002/apj.1892. What a probable mechanism of the boron introduction into C- and N-containing phase?
- all figures are to small and should be enlarged;
- in Str. 301 authors claim that B is capable to brake Fe-O bond that seems to point on Fe-B bond formation. However, this fact has not been confirmed in the study;
- in str. 339-340 authors mention that P-Fe@CN and B-Fe@CN displayed decreased degradation performance, while in Str. 335 B-Fe@CN is the second by its efficiency.

Comments on the Quality of English Language

The English requires some minor corrections

Reviewer 5 Report

Comments and Suggestions for Authors

MS ID: water-2725485

Title: Multi-heteroatom doped Fe@CN activation PMS for the degradation of trace organic contaminants from water: fabrication optimization and performance 

Comments: 

The current manuscript is not found suitable for publication in its present form. The authors may refer to the points below for improvement of the manuscript.

1-The quality of English could be better in many sentences of this manuscript, with basic grammatical mistakes and, notably, poor framing of the sentences in many places. The MS should be proofread by a native speaker or a professional in the field. 

2-Please revise the Title; a coma is missing after the word 'fabrication.' 

3-Abstract needs to be substantially improved. This section must be started with the justification of this study. 

4-Please carefully revise all the formulas and units used in this study. 

5-A section of nomenclature/abbreviations is suggested. 

6-Some figures are poor in quality and hard to follow.  

Comments on the Quality of English Language

Moderate editing of English language required

Round 2

Reviewer 2 Report

Comments and Suggestions for Authors

I appreciate the authors for revising the manuscript and addressing my comments. However, I still recommend that the authors separate the pure sorption and degradation results and discuss them separately. In Figure 8e, an instantaneous drop in ATL concentration within the initial 5 minutes should be attributed to pure sorption with a B-Fe@CN concentration of 100 mg/L, resulting in >40% ATL removal. Similarly, in Figure 8f, >60% of ALT was instantly adsorbed with PB-Fe@CN at 600 mg/L. The same applies to the case of PMS=1.3mM in Figure 8i. However, the authors attributed them to degradation and used them to calculate the degradation rate, which is not correct.

I can reconsider the acceptance of the manuscript once the authors address my comments.

Round 3

Reviewer 2 Report

Comments and Suggestions for Authors

The authors have addressed my comments. The manuscript can be accepted now.